# Generalized Variational Inference in Function Spaces: Gaussian Measures meet Bayesian Deep Learning

**Veit D. Wild**[*]
Department of Statistics
University of Oxford
veit.wild@stats.ox.ac.uk

**Robert Hu**[* †]
Amazon
robyhu@amazon.co.uk

**Dino Sejdinovic** [†]
School of Computer and Mathematical Sciences
University of Adelaide
dino.sejdinovic@adelaide.edu.au

## Abstract

We develop a framework for generalized variational inference in infinite-dimensional function spaces and use it to construct a method termed Gaussian Wasserstein inference (GWI). GWI leverages the Wasserstein distance between Gaussian measures on the Hilbert space of square-integrable functions in order to determine a variational posterior using a tractable optimization criterion. It avoids pathologies arising in standard variational function space inference. An exciting application of GWI is the ability to use deep neural networks in the variational parametrization of GWI, combining their superior predictive performance with the principled uncertainty quantification analogous to that of Gaussian processes. The proposed method obtains state-of-the-art performance on several benchmark datasets.

## 1 Introduction

In the past decade, considerable effort has been invested in developing Bayesian deep learning approaches [Welling and Teh, 2011, Chen et al., 2014, Blundell et al., 2015, Gal and Ghahramani, 2016, Kendall and Gal, 2017, Ritter et al., 2018, Khan et al., 2018, Maddox et al., 2019]. There are at least two key advantages to Bayesian models. Firstly, Bayesian model averaging is known to improve predictive performance [Komaki, 1996] even in misspecified situations [Fushiki, 2005, Ramamoorthi et al., 2015]. The empirical success of methods such as deep ensembles [Lakshminarayanan et al., 2017] may be interpreted as compelling evidence for this claim [Wilson and Izmailov, 2020]. Secondly, Bayesian models provide the user with a predictive distribution for an unseen data point. This can be naturally leveraged to quantify posterior uncertainty.

Even though impressive progress has been made, there are problems that remain unresolved. The prior distribution for the unknown function is typically induced by a prior distribution over deep neural network weights (and biases). It is hard to interpret the inductive bias in a function space that is induced by such priors for weights and unclear how one might incorporate prior knowledge about the unknown function. Additionally, the resulting inference problem is extremely high-dimensional and requires approximation techniques that are either computationally expensive [Neal, 2012] or so crude that the approximate posterior may suffer from pathological behavior [Foong et al., 2020]. The difficulties of performing Bayesian inference for weights have led to the emergence of methods that

---

[*]equal contribution, order decided by coinflip
[†]Work mainly done while the authors were with the Department of Statistics, University of Oxford

36th Conference on Neural Information Processing Systems (NeurIPS 2022).

approach the problem in function space directly [Ma et al., 2019, Sun et al., 2019, Rudner et al., 2020, Ma and Hernández-Lobato, 2021].

The theory of constructing prior distributions in function spaces is well developed and the most famous class of prior distributions are *Gaussian processes*. They have been commonly used for decades in the machine learning community to elicit interpretable functional priors and are known to have well-calibrated predictive uncertainties [Rasmussen, 2003].

In a separate thread of research, a new powerful inference framework called *Generalized Variational Inference* (GVI) has been recently developed [Knoblauch et al., 2019]. The authors argue that standard assumptions of Bayesian inference such as well-specified priors, well-specified likelihoods and infinite computing power are often violated in practice. They therefore propose a generalized view on Bayesian inference that takes these points into consideration. We extend the work of Knoblauch et al. [2019] to situations where no probability density functions for the prior exist and are thus able to use generalized variational inference in infinite-dimensional function spaces directly. We then specify both the prior and variational measures as Gaussian measures and measure their dissimilarity using the Wasserstein distance. This results in the method which we call *Gaussian Wasserstein Inference in Function Spaces* (GWI-FS). An exciting application of our method is the ability to equip deep neural networks with uncertainty quantification using the framework analogous to that of Gaussian processes, resulting in a state-of-the-art method termed *GWI-net*. Our main contributions are:

- We create a general framework for inference in function space based on Gaussian measures on the space of square-integrable functions,
- We derive an objective function that can be expressed in terms of the *parameters of the Gaussian measures*,
- We derive a tractable approximation to our objective function that is valid for (almost) arbitrary kernels and mean functions,
- We demonstrate the utility of our method by obtaining state-of-the-art results on the UCI regression datasets and on Fashion MNIST and CIFAR 10[3].

## 2 Related Work

GWI-FS draws on the work developed in the Gaussian process literature, but can be used to equip traditional neural network architectures with uncertainty. We therefore give a brief overview of the relevant related methods in both the Bayesian neural network (BNNs) and Gaussian process community.

**Bayesian neural networks** Traditionally Bayesian neural networks have been assigned priors in weight space. The effects of various priors on inference and uncertainty quantification are still not well understood [Fortuin et al., 2021]. As the posterior (over weights) is intractable, sampling algorithms such as Hamiltonian Monte Carlo (HMC) were initially proposed Neal [2012]. Due to the unfavorable scaling properties of standard HMC which requires the full gradient, batch-size approximations of HMC evolved [Chen et al., 2014]. Another line of research exploits Langevin dynamics to generate posterior samples [Welling and Teh, 2011] in weight space.

**Variational methods for BNNs in weight space** In variational inference, the true posterior is approximated by a more tractable so-called *variational* distribution. The user specifies a class of approximate posterior measures and selects the best posterior approximation by maximizing the so-called evidence lower bound (ELBO). The Bayes by Backprop [Blundell et al., 2015] method is one such variational mean-field approximation of the weight-space posterior. In variational dropout [Gal and Ghahramani, 2016], a specific approximation is chosen to reinterpret dropout [Srivastava et al., 2014] at test time as a variational procedure.

**Variational methods for BNNs in function spaces** Inference in weight space is challenging, as the problem is typically high-dimensional and the posterior distribution over weights multi-modal. This led to a line of research in which inference algorithms are formulated in function spaces. Variational implicit processes [Ma et al., 2019] approximate the BNN posterior as a linear combination of draws from the prior. Functional-BNN [Sun et al., 2019] matches a BNN to a functional prior (for example a GP) and performs inference by optimising a functional Kullback-Leibler (KL) divergence exploiting score function estimators [Li and Turner, 2017, Shi et al., 2018]. Rudner et al. [2020] use a local approximation to the prior and variational posterior processes to obtain a tractable functional

---

[3]Codebase: https://github.com/MrHuff/GWI

Kullback-Leibler divergence. Ma and Hernández-Lobato [2021] generalise the variational family in Ma et al. [2019] and obtain a more scalable procedure by using a different approximation to the functional KL-divergence. Recent work has also proposed to adapt BNN priors to interpretable functional priors by minimizing the Wasserstein distance between a BNN prior and a Gaussian process [Tran et al., 2020]. Another line of research exploits the Wasserstein gradient flow and tries to encourage diversity in the function space [D'Angelo et al., 2021, D'Angelo and Fortuin, 2021].

**Gaussian processes** Standard Gaussian process regression [Rasmussen, 2003] allows interpretable prior specification but scales poorly with respect to the number of data points. As a result, a plethora of approximation techniques are introduced. On one hand, there are variational approximations to the true posterior [Titsias, 2009, Hensman et al., 2013] and several extensions [Hensman et al., 2017, Salimbeni et al., 2018, Dutordoir et al., 2020]. On the other hand, GPU utilization is combined with Krylov subspace methods to obtain scalability [Gardner et al., 2018, Wang et al., 2019].

## 3 Background

In this section we give some background on generalized variational inference in infinite dimensions and introduce Gaussian measures in Hilbert spaces. We further discuss their relation to the more familiar Gaussian processes at the end.

### 3.1 Generalized Variational Inference in Function Spaces

In functional variational inference, we assign a prior $p(f)$ to the unknown function $f \in E$, where $E$ is a function space[4]. The prior is combined with the likelihood $p(y|f)$ to give the posterior $p(f|y)$. The posterior is often intractable which is why in variational inference we specify a tractable variational approximation $q(f)$ to $p(f|y)$ and train our model by maximising the evidence lower bound (ELBO)

$$\mathcal{L} = \mathbb{E}_{q(f)}\big[\log p(y|f)\big] - \mathbb{D}_{\mathrm{KL}}\big(q(f), p(f)\big), \tag{1}$$

where $\mathbb{D}_{\mathrm{KL}}$ denotes the KL divergence. Note that in the case where $E$ is infinite dimensional $p(f)$ and $q(f)$ cannot be probability density functions with respect to the Lebesgue measure [see e.g. Hunt et al., 1992, for a discussion], which is why the above notation, although commonly used, is imprecise. What we in fact mean are the probability measures over $E$ associated with the prior and variational approximation. We will denote these measures as $\mathbb{P}^F$ and $\mathbb{Q}^F$ from now on to make this difference explicit. The ELBO in this notation reads as

$$\mathcal{L} := \mathbb{E}_{\mathbb{Q}}\big[\log p(y|F)\big] - \mathbb{D}_{\mathrm{KL}}\big(\mathbb{Q}^F, \mathbb{P}^F\big). \tag{2}$$

Note that the KL divergence (for measures) is defined as

$$\mathbb{D}_{\mathrm{KL}}\big(\mathbb{Q}^F, \mathbb{P}^F\big) = \int \log\left(\frac{d\mathbb{Q}^F}{d\mathbb{P}^F}(f)\right) d\mathbb{Q}^F(f), \tag{3}$$

where we assume that $\mathbb{Q}^F$ is dominated by the measure $\mathbb{P}^F$ which guarantees the existence of the Radon-Nikodym derivative $d\mathbb{Q}^F/d\mathbb{P}^F$. A number of papers focus on obtaining tractable approximations of (3) [Sun et al., 2019, Rudner et al., 2020, Ma and Hernández-Lobato, 2021]. However, the use of KL-divergence in infinite-dimensional function spaces can be a delicate task, since benign constructions of priors and variational approximations may not satisfy that $\mathbb{Q}^F$ is dominated by $\mathbb{P}^F$ which leads to $\mathbb{D}_{\mathrm{KL}}\big(\mathbb{Q}^F, \mathbb{P}^F\big) = \infty$ [Burt et al., 2020]. This often renders the objective (2) useless or at least problematic.

A *true Bayesian* is committed to the use of the KL divergence in (2) as maximizing $\mathcal{L}$ is equivalent to minimizing the KL divergence between the true posterior measure and the variational measure. This equivalence is typically demonstrated using pdfs but the argument generalizes to infinite dimensions as is shown for GPs in Matthews et al. [2016] or in a more measure theoretic formulation in Theorem 4 of Wild and Wynne [2021].

However, Knoblauch et al. [2019] argue that given the problems of prior and likelihood specification as well as available compute, an axiomatically justified way of moving from prior to posterior beliefs is by solving a more general optimization problem [Knoblauch et al., 2019, Theorem 15]. Crucially it is valid to replace the KL-divergence by an arbitrary measure of dissimilarity $\mathbb{D}$ satisfying $\mathbb{D}(\mathbb{Q}^F, \mathbb{P}^F) \geq 0$ and $\mathbb{D}\big(\mathbb{Q}^F, \mathbb{P}^F\big) = 0 \Rightarrow \mathbb{Q}^F = \mathbb{P}^F$. The arguments in Knoblauch et al. [2019] are

---

[4]We assume $E$ to be a Polish space, which avoids technical difficulties in defining the posterior measure [Ghosal and Van der Vaart, 2017, Chapter 1.3 ]

made assuming the existence of a pdf for the prior, but they rely solely on a reformulation of Bayesian inference as optimization problem [Knoblauch et al., 2019, Chapter 2]. We show in Appendix A.1 that this reformulation can also be made for infinite-dimensional prior measures and therefore consider the generalized loss

$$\mathcal{L} := -\mathbb{E}_{\mathbb{Q}}\big[\log p(y|F)\big] + \mathbb{D}\big(\mathbb{Q}^F, \mathbb{P}^F\big), \tag{4}$$

a valid optimization objective for an arbitrary dissimilarity measure $\mathbb{D}$. This is merely an infinite-dimensional version of equation (10) in Knoblauch et al. [2019]. We refer to inference targeting the objective (4) as *Generalized variational inference in function space* (GVI-FS).

Generalised variational inference can be interpreted as regularised loss minimisation lifted into the space of probability measures. The first term in (4) is understood as a loss which we want to minimise on average, while the second term punishes strong deviations from the prior.

The particular instance of GVI-FS that we explore is where both $\mathbb{P}^F$ and $\mathbb{Q}^F$ are Gaussian measures (on an infinite-dimensional Hilbert space) and $\mathbb{D}$ is chosen to be the Wasserstein metric [Kantorovich, 1960]. We will refer to this setting as *Gaussian Wasserstein Inference in Function Space* (GWI-FS) or more consciously as *Gaussian Wasserstein Inference* (GWI)

## 3.2   Gaussian Random Elements and Gaussian Measures in Hilbert spaces

In this section we introduce Gaussian random elements (GRE) and Gaussian measures in Hilbert spaces – these concepts are somewhat technical but crucial in the construction of our method. We then describe their close relationship to the more familiar Gaussian process notions in the next section.

Let $\big(\Omega, \mathcal{A}, \mathbb{P}\big)$ be the underlying (physical) probability space and $\big(H, \langle \cdot, \cdot \rangle\big)$ be a Hilbert space.

**Gaussian random elements**   A measurable function $F : \Omega \to H$ is called GRE (in $H$) if and only if $\langle F, h \rangle : \Omega \to \mathbb{R}$ has a scalar Gaussian distribution for all $h \in H$.[5] Every GRE $F$ has a mean element $m \in H$ defined by

$$m := \int F(\omega)\, d\mathbb{P}(\omega) \tag{5}$$

and a (linear) covariance operator $C : H \to H$ defined by

$$Ch(\cdot) := \int \langle F(\omega), h \rangle F(\omega)\, \mathbb{P}(\omega) - \langle m, h \rangle m. \tag{6}$$

for $h \in H$. Both integrals are to be understood as Bochner integrals [Kukush, 2020, Chapter 3]. The Bochner integral has the property that $\big\langle \int F(\omega)\, d\mathbb{P}(\omega), h \big\rangle = \int \langle F(\omega), h \rangle\, d\mathbb{P}(\omega)$ for all $h \in H$. This combined with Fubini's theorem and the definition of a GRE implies that

$$\langle F, h \rangle \sim \mathcal{N}(\langle m, h \rangle, \langle Ch, h \rangle), \tag{7}$$

for any $h \in H$ with $\mathcal{N}(\mu, \sigma^2)$ denoting the normal distribution with mean $\mu \in \mathbb{R}$ and variance $\sigma^2 > 0$. Similarly we denote $F \sim \mathcal{N}(m, C)$ for a GRE in $H$ with mean element $m$ and covariance operator $C$. It can be shown that the covariance operator $C$ of a GRE is a positive self-adjoint trace-class operator. Conversely, for every positive self-adjoint trace class operator and every $m \in H$, there exists a GRE with $F \sim \mathcal{N}(m, C)$ [Bogachev, 1998, Theorem 2.3.1].

**Gaussian measures**   The push-forward measure of $\mathbb{P}$ through $F$ is defined as $\mathbb{P}^F(A) := \mathbb{P}\big(F^{-1}(A)\big)$ for all Borel-measurable $A \subset H$. If $F \sim \mathcal{N}(m, C)$ is a GRE, we call $P := \mathbb{P}^F$ a GM and write $P = \mathcal{N}(m, C)$. Note that GMs or equivalently GREs allow us to specify probability distributions over (infinite-dimensional) Hilbert spaces by using a given mean element and a given covariance operator.

Details about Gaussian Measures in Hilbert spaces can be found in Chapter 2 of Da Prato and Zabczyk [2014] or in Kukush [2020]. In fact, Gaussian measures can be defined on even more general linear spaces such as Banach or Fréchet spaces [Bogachev, 1998].

## 3.3   Gaussian Processes and Their Corresponding Measures

In this section we describe how Gaussian processes – a standard tool to assign functional priors in Bayesian machine learning – are related to Gaussian measures.

---

[5]We allow for the degenerate case where the variance of $\langle F, h \rangle$ is zero. This means we interpret a Gaussian with variance zero as Dirac measure.

Let $(\Omega, \mathcal{A}, \mathbb{P})$ be the underlying (physical) probability space and $\mathcal{X} \subset \mathbb{R}^D$ be measurable. The (product-) measurable mapping $G : \Omega \times \mathcal{X} \to \mathbb{R}$ is called a Gaussian process (GP) if and only if for all $N \in \mathbb{N}$ and all $X = \{x_n\}_{n=1}^N \subset \mathcal{X}$ the random vector $G(X) := \left(G(\cdot, x_1), \ldots, G(\cdot, x_N)\right)^T$ is multivariate Gaussian. For a GP $G$ we define a mean function $m(x) := \mathbb{E}\left[G(x)\right]$, $x \in \mathcal{X}$, and a covariance function by $k(x, x') := \mathbb{C}\left[G(x), G(x')\right]$ for $x, x' \in \mathcal{X}$. Here $\mathbb{E}$ denotes the expected value and $\mathbb{C}[\cdot, \cdot]$ the covariance. It follows from the definition that $G(X) \sim \mathcal{N}\left(m(X), k(X, X)\right)$ for any $\{x_n\}_{n=1}^N \subset \mathcal{X}$, where we define $m(X) := \left(m(x_n)\right)_{n=1}^N$ and $k(X, X) := \left(k(x_n, x_{n'})\right)_{n,n'=1}^N$. We write $G \sim GP(m, k)$ for a GP with mean function $m$ and covariance function $k$. Note that by the properties of the covariance we know that $k(X, X)$ is a (symmetric) positive semi-definite matrix for all $\{x_n\}_{n=1}^N \subset \mathcal{X}$ and $N \in \mathbb{N}$. A function with this property is called *kernel*, a terminology that we adopt henceforth. Kolmogorov's existence theorem [Billingsley, 2008, Section 36] guarantees the existence of a Gaussian process for any kernel $k$ and any mean function $m$. The standard reference for Gaussian processes in machine learning is Rasmussen [2003].

The main advantage of Gaussian processes in specifying priors over a function space is that the kernel $k$ allows us to incorporate readily interpretable prior assumptions, such as smoothness or periodicity. For example, choosing the squared exponential kernel [Rasmussen, 2003] implies that the unknown function is infinitely differentiable and that the correlation of the functional output is higher the closer the inputs are.

In order to insert the Gaussian process prior into our generalized loss in (4) we need to know the probability measure that is associated to the Gaussian process. In general, we can associate more than one Gaussian measure with a given Gaussian process. For example:

- If the GP has continuous sample paths we can associate a Gaussian measure on the space $E$ of continuous functions with it [Lifshits, 2012, Example 2.4].
- If the GP has square-integrable sample paths we can associate a Gaussian measure on the Hilbert space of square-integrable functions with it (cf. Theorem 1).

These sample path properties can be guaranteed under additional assumptions on the kernel. The next theorem discusses one such kernel condition which guarantees the GP to have sample paths in the Hilbert space of square integrable functions, denoted $L^2(\mathcal{X}, \rho, \mathbb{R})$, with inner product $\langle g, h \rangle_2 := \int_{\mathcal{X}} g(x)h(x)\, d\rho(x)$.

**Theorem 1.** *Let $F \sim GP(m, k)$ be a GP with mean $m \in L^2(\mathcal{X}, \rho, \mathbb{R})$ and kernel $k$ such that*

$$\int_{\mathcal{X}} k(x, x)\, d\rho(x) < \infty. \tag{8}$$

*We call a kernel satisfying (8) trace-class kernel. Then the mapping $\widetilde{F} : \Omega \to L^2(\mathcal{X}, \rho, \mathbb{R})$ defined as $\widetilde{F}(\omega) := F(\omega, \cdot)$ is a Gaussian random element with mean $m$ and covariance operator $C$ given as*

$$Cg(\cdot) := \int k(\cdot, x')g(x')\, d\rho(x') \tag{9}$$

*for any $g \in L^2(\mathcal{X}, \rho, \mathbb{R})$. Consequently $P := \mathbb{P}^F \sim \mathcal{N}(m, C)$ is a Gaussian measure.*

*Proof.* The fact that $\widetilde{F}$ as defined above is a GRE follows immediately from Example 2.3.16 in Bogachev [1998]. The fact that $m$ is its mean and $C$ as defined in (9) is its covariance operator follows from Fubini's theorem. $\square$

It shall be noted that there is no need to appeal to GPs in order to justify the use of GMs. In fact, it has recently been demonstrated that variational inference for GPs can be formulated purely in terms of GMs [Wild and Wynne, 2021]. In the following sections we will therefore deploy GMs without any reference to GPs, but it is of course always possible to think of them as the measures that correspond to GPs where the kernel satisfies an additional assumption such as (8).

## 4 Gaussian Wasserstein Inference in Function Spaces

This section describes how the Wasserstein distance between Gaussian measures can be used to obtain a tractable optimization target for inference in function spaces. In the end, we discuss several parametrizations of GWI and introduce our main inference method - the GWI-net.

## 4.1 Model description

Let $\{(x_n, y_n)\}_{n=1}^N \subset \mathcal{X} \times \mathcal{Y}$ be $N \in \mathbb{N}$ paired observations. We assume that $\mathcal{X} \subset \mathbb{R}^D$, $D \in \mathbb{N}$ and further that $\mathcal{Y} = \mathbb{R}$ for regression and $\mathcal{Y} = \{1, \ldots, J\}$ for classification with $J \in \mathbb{N}$ classes. We focus in our exposition here on the regression case but have given the relevant derivations for classification in Appendix A.6.

As pointed out in section 3.1, GVI in function space minimises the generalized loss $\mathcal{L} = -\mathbb{E}_{\mathbb{Q}}\big[\log p(y|F)\big] + \mathbb{D}(\mathbb{Q}^F, \mathbb{P}^F)$. We make the mild assumption that the unknown function $f$ is square integrable with respect to the data distribution $\rho$ on $\mathcal{X}$ which means $f \in E = L^2(\mathcal{X}, \rho, \mathbb{R})$. The prior $P := \mathbb{P}^F$ is described by a Gaussian measure with mean $m_P \in \mathcal{L}^2(\mathcal{X}, \rho, \mathbb{R})$ and covariance operator $C_P$ described by a trace-class kernel $k : \mathcal{X} \times \mathcal{X} \to \mathbb{R}$ which means it is given as $(C_P f)(x) := \int_{\mathcal{X}} k(x, x') f(x') \, d\rho(x')$ for all $f \in L^2(\mathcal{X}, \rho, \mathbb{R})$. We assume a Gaussian likelihood for $y := (y_1, \ldots, y_N)$ given as $p(y|f) := \prod_{n=1}^N p(y_n|f)^6$ with

$$p(y_n|f) := \mathcal{N}(y_n \mid f(x_n), \sigma^2), \tag{10}$$

where $\mathcal{N}(\cdot \mid \mu, \sigma^2)$ denotes the pdf of a normal distribution with mean $\mu \in \mathbb{R}$ and variance $\sigma^2 > 0$. This prior and likelihood are natural choices as they mimic the standard formulation of Gaussian process regression. The variational approximation of the posterior is chosen to be another Gaussian measure $Q := \mathbb{Q}^F$ with arbitrary mean $m_Q \in L^2(\mathcal{X}, \rho, \mathbb{R})$ and arbitrary covariance operator $C_Q$ induced by a trace-class kernel $r$: $(C_Q f)(x) := \int_{\mathcal{X}} r(x, x') f(x') \, d\rho(x')$ for all $f \in L^2(\mathcal{X}, \rho, \mathbb{R})$.

It remains for us to select a dissimilarity measure $\mathbb{D}$. As already pointed out in the introduction we decide to use the Wasserstein distance $W_2$ (a formal definition is given in Appendix A.3). This choice was guided by two considerations:

1. The Wasserstein metric was proven to be a useful metric for probability distributions in machine learning applications [Arjovsky et al., 2017, Tran et al., 2020]. Furthermore the Wasserstein metric is known to have desirable statistical properties [Panaretos and Zemel, 2019].

2. The Wasserstein distance is tractable for arbitrary Gaussian measures on (separable) Hilbert spaces [Gelbrich, 1990] and given as

$$W_2^2(P, Q) = \|m_P - m_Q\|_2^2 + tr(C_P) + tr(C_Q) - 2 \cdot tr\Big[\big(C_P^{1/2} C_Q C_P^{1/2}\big)^{1/2}\Big], \tag{11}$$

   where $tr$ denotes the trace of an operator and $C_P^{1/2}$ is the square root of the positive, self-adjoint operator $C_P$. This is in stark contrast to the KL-divergence that is infinite whenever $\mathbb{Q}^F$ is not dominated by $\mathbb{P}^F$ and even in the case where it is finite there exists no explicit formula for the KL-divergence in infinite dimensions.

The generalized loss for our model is therefore given as

$$\mathcal{L} = -\sum_{n=1}^N \mathbb{E}_{\mathbb{Q}}\Big[\log \mathcal{N}\big(y_n \mid F(x_n), \sigma^2\big)\Big] + W_2(P, Q). \tag{12}$$

Note that the expected log-likelihood in (12) can be calculated analytically as

$$\mathbb{E}_{\mathbb{Q}}\Big[\log \mathcal{N}\big(y_n \mid F(x_n), \sigma^2\big)\Big] = -\frac{N}{2} \log(2\pi\sigma^2) - \sum_{n=1}^N \frac{\big(y_n - m_Q(x_n)\big)^2 + r(x_n, x_n)}{2\sigma^2}. \tag{13}$$

It remains to produce an approximation of (11) in order to obtain a tractable inference procedure. To this end, note that by definition $\|m_P - m_Q\|_2^2 = \int \big(m_P(x) - m_Q(x)\big)^2 d\rho(x)$ and further $tr(C_P) = \int k(x, x) \, d\rho(x)$ [Brislawn, 1991]. We now replace the true input distribution $\rho$ with the empirical data distribution $\widehat{\rho} := \frac{1}{N} \sum_{n=1}^N \delta_{x_n}$, where $\delta_x$ denotes the Dirac measure in $x \in \mathcal{X}$. This gives $\|m_P - m_Q\|_2^2 \approx \frac{1}{N} \sum_{n=1}^N \big(m_P(x_n) - m_Q(x_n)\big)^2$, $tr(C_P) \approx \frac{1}{N} \sum_{n=1}^N k(x_n, x_n)$ and

---

[6]Astute readers may notice that the definition of the likelihood contains a pointwise evaluation $f(x_n)$ which may not be a well defined operation on $L^2(\mathcal{X}, \rho, \mathbb{R})$. We detail in Appendix 30 how that problem can be circumvented and that in fact $F(x) \sim \mathcal{N}(m(x), k(x, x))$ as one would expect.

$tr(C_Q) \approx \frac{1}{N} \sum_{n=1}^{N} r(x_n, x_n)$. It remains to provide an approximation of $tr\left[\left(C_P^{1/2} C_Q C_P^{1/2}\right)^{1/2}\right]$.
The key idea is to approximate the spectrum of $C_P^{1/2} C_Q C_P^{1/2}$ by that of an appropriate kernel matrix. Details are discussed in Appendix A.4. This leads to the following final approximation for the Wasserstein metric

$$\hat{W}^2 := \frac{1}{N} \sum_{n=1}^{N} \left(m_P(x_n) - m_Q(x_n)\right)^2 + \frac{1}{N} \sum_{n=1}^{N} k(x_n, x_n) \tag{14}$$

$$+ \frac{1}{N} \sum_{n=1}^{N} r(x_n, x_n) - \frac{2}{\sqrt{NN_S}} \sum_{s=1}^{N_S} \sqrt{\lambda_s\left(r(X_S, X)k(X, X_S)\right)}, \tag{15}$$

where $X_S := (x_{S,1}, \ldots, x_{S,N_S})$ with $x_{S,1}, \ldots x_{S,N_S} \in \mathbb{R}^D$ being subsampled from the input data $X$. Further $r(X_S, X) := \left(r(x_{S,s}, x_n)\right)_{s,n}$ and $k(X, X_S) := \left(k(x_n, x_{S,s})\right)_{n,s}$ for $n = 1, \ldots, N$, $s = 1, \ldots, N_S$ and $\lambda_s\left(r(X_S, X)k(X, X_S)\right)$ denotes the $s$-th eigenvalue of the matrix $r(X_S, X)k(X, X_S) \in \mathbb{R}^{N_S \times N_S}$. The approximation quality of $\widehat{W}$ is related to the spectral decay of the operator $C_P C_Q$, which in turn is determined by the kernels $k$ and $r$. For the choices made in Section 4.2 we empirically observe rapid spectral decay (cp. Appendix A.13) and therefore are confident that the 2-Wasserstein distance is estimated reliably for our method.

The combination of (13), (14) and (15) gives a generalized loss that is tractable in terms of $m_P, m_Q, k$, and $r$. If we disregard computation time of $m_P, m_Q, k$ and $r$, the generalized loss can be evaluated in $\mathcal{O}(N + N_S^2 N + N_S^3)$, where typically $N_S \ll N$, e.g. $N_S = 100$. We provide a batch version of our loss in Appendix A.5 which reduces the computations to $\mathcal{O}(N_S^2 N_B + N_S^3)$ where $N_B \ll N$ is the batch-size. Note, however, that the final computation time for our method will be determined by the complexity hidden in the evaluation of $m_Q, m_P, k$, and $r$ as we need $N_B$ evaluations of $m_Q$ and $m_P$ and $N_S \cdot N_B$ evaluations of $r$ and $k$ per iteration.

## 4.2 Parameterisations of Prior and Variational Measure

The prior for our model is given as $P = \mathcal{N}(m_P, C_P)$ with $C_P$ induced by a trace-class kernel $k$. One of the advantages of the proposed approach is that any trace-class kernel is allowed and this is where one can incorporate specific assumptions and domain expertise. This is a thoroughly studied topic: the prior kernel can encode periodicity [Durrande et al., 2016], geometric intuition [van der Wilk et al., 2018], and even model linear constraints for the unknown function [Jidling et al., 2017]. In order to keep the exposition simple and maintain focus on the inference, however, and in line with using simple priors on network weights in standard Bayesian deep learning, we opt for a simple zero mean prior $m_P = 0$ and a standard ARD kernel $k$ given as

$$k(x, x') = \sigma_f^2 \exp\left(-\frac{1}{2} \sum_{d=1}^{D} \frac{(x_d - x_d')^2}{\alpha_d^2}\right) \tag{16}$$

for $x, x' \in \mathcal{X} \subset \mathbb{R}^D$. We refer to $\sigma_f > 0$ as *kernel scaling factor* and to $\alpha_d > 0$ as *length-scale* for dimension $d$. The parameters $\sigma_f$ and $\alpha := (\alpha_1, \ldots, \alpha_D)$ are called *prior hyperparameters*.

The rest of the section explores various choices for the variational mean $m_Q$ and the variational kernel $r$. The parameters appearing in the specification of $m_Q$ and $r$ are referred to as *variational parameters*.

**GWI: Stochastic variational Gaussian process**     Let $z_1, \ldots, z_M \in \mathcal{X}$ be a subsample of the data $X$ with $M \ll N$. We define the posterior mean

$$m_Q(x) := m_P(x) + \sum_{m=1}^{M} \beta_m k_m(x) \tag{17}$$

with $\beta_m \in \mathbb{R}$ and $k_m(x) := k(x, z_m)$, $m = 1, \ldots, M$ where $k$ is the prior kernel $k$ and $\beta := (\beta_1, \ldots, \beta_M) \in \mathbb{R}^M$ are variational parameters. Define further the variational kernel

$$r(x, x') = k(x, x') - k_Z(x)^T k(Z, Z)^{-1} k_Z(x) + k_Z(x)^T \Sigma k_Z(x), \tag{18}$$

where $\Sigma \in \mathbb{R}^{M \times M}$ is the symmetric and positive definite variational covariance matrix that parameterises $r$. This choice of $m_Q$ and $r$ essentially recovers the *stochastic variational Gaussian processes*

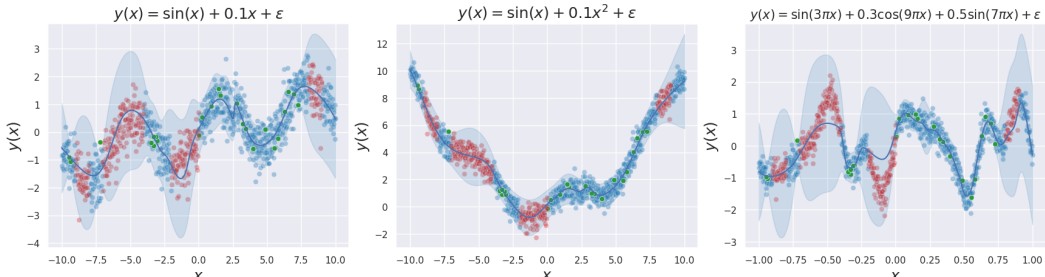

Figure 1: ■ : Training data    ■ : Unseen data    ■ : Inducing points
We query the above functions at $N = 1000$ equidistant points and add white noise with $\epsilon \sim \mathcal{N}(0, 0.5^2)$. We use $M = 30$ inducing points and train our method as described in Appendix A.7. The plot shows $m_Q(x) \pm 1.96\sqrt{\mathbb{V}[Y^*(x)|Y]}$ where $\mathbb{V}[Y^*(x)|Y]$ is the posterior predictive variance given as $r(x,x) + \sigma^2$.

(SVGP) model of Titsias [2009]. Note that in our framework it is straightforward to use all (or just more) basis functions for the mean $m_Q(x) := m_P(x) + \sum_{n=1}^{N} \beta_n k_n(x)$ where $k_n(x) := k(x, x_n)$, $\beta_n \in \mathbb{R}$, $n = 1, \ldots, N$. This mirrors the construction in Cheng and Boots [2017] where we allow more parameters to learn the mean than in SVGP. However, both Titsias [2009] and Cheng and Boots [2017] use a different objective function than GWI to learn the unknown parameters.

**GWI: deep neural network with SVGP**    An interesting approach is to parameterise the posterior mean as a deep neural network (DNN). We assume the DNN has $L \in \mathbb{N}$ hidden layers and the width of layer $\ell = 1, \ldots, L$ is denoted $D_\ell$ with $D_0 := D$ and $D_{L+1} = 1$. This means we define $g^1(x) := W^1 x + b^1$ and further $h^\ell(x) := \phi(g^\ell(x))$, $g^{\ell+1}(x) := W^{\ell+1} h^\ell(x) + b^{\ell+1}$ for $\ell = 1, \ldots, L$. Here $W^{l+1}$ is $D_{\ell+1} \times D_\ell$ matrix, $b^{\ell+1} \in \mathbb{R}^{D_{\ell+1}}$ is a bias vector for layer $l$ and $\phi$ an activation function. We can then define the variational mean as $m_Q(x) := m_P(x) + g^{L+1}(x)$. If we choose the SVGP kernel $r$ in (18), we essentially predict with a neural network and quantify uncertainty with a (sparse) Gaussian process, capturing the beneficial properties of both.

Neural networks have been combined in several ways with GPs [Wilson et al., 2016, Tran et al., 2020]. However, to the best of our knowledge they were not used to directly parametrize the posterior in the context of generalized variational inference in function space. The spirit of our approach is fundamentally different: rather than thinking of a neural network as a model which needs to be made Bayesian, we use it as a parametrisation of a variational posterior.

We note that we do not here provide an exhaustive study on how to best parameterize the variational measure. This paper is focused on demonstrating the ability of the proposed method to obtain valid uncertainty quantification. An exploratory study on how properties and quality of uncertainty quantification relate to different choices of $m_Q$ and $r$ is reserved for future work. We mention potential problems that can occur from misspecification in Appendix A.10.

## 5    Experiments

We show results for GWI with the SVGP mean (17) and the SVGP kernel (18). We use the shorthand GWI: SVGP for this approach. Additionally we implement the DNN mean with the SVGP kernel (18). This combination achieves impressive results on various regression and classification tasks. We call this method GWI: DNN-SVGP or simply GWI-net.

**Illustrative Examples**    In Figure 1 we illustrate GWI-net on a few toy examples. One can clearly see that the posterior predictive variance expands for regions lacking observations which demonstrates the ability of our method to quantify uncertainty. We provide an additional graphic comparison with SVGP in Appendix A.12 and an example for two-dimensional inputs in Appendix A.9

There we show that the pathologies regarding the quantification of in-between uncertainty discussed in Foong et al. [2020] are not present for our method.

**UCI Regression**    In Table 1 we report the average test negative log-likelihood (NLL) (cf. Appendix A.7 for details) of GWI: SVGP and GWI-net (GWI: DNN-SVGP) and the results of several weight-space approaches for BNNs: Bayes-by-Backprop (BBB) [Blundell et al., 2015], variational dropout

(VDO) [Gal and Ghahramani, 2016], and variational alpha dropout ($\alpha = 0.5$) [Li and Gal, 2017]. We also compare with four function-space BNN inference methods: functional variational inference with BNN prior (FVI) [Ma and Hernández-Lobato, 2021], variationally implicit processes (VIP) with BNNs, VIP-Neural processes [Ma et al., 2019], and functional BNNs (FBNNs) [Sun et al., 2019]. In order to ensure a fair comparison we matched neural network architectures and training procedures for the different methods. Detailed explanations are given in Appendix A.7.

| Dataset | N | D | GWI | | FVI | VIP-BNN | VIP-NP | BBB | VDO | $\alpha = 0.5$ | FBNN | **EXACT GP** |
|---|---|---|---|---|---|---|---|---|---|---|---|---|
| | | | SVGP | DNN-SVGP | | | | | | | | |
| BOSTON | 506 | 13 | 2.8±0.31 | **2.27±0.06** | 2.33±0.04 | 2.45±0.04 | 2.45±0.03 | 2.76±0.04 | 2.63±0.10 | 2.45±0.02 | 2.30±0.10 | 2.46±0.04 |
| CONCRETE | 1030 | 8 | 3.24±0.09 | **2.64±0.06** | 2.88±0.06 | 3.02±0.02 | 3.13±0.02 | 3.28±0.01 | 3.23±0.01 | 3.06±0.03 | 3.09±0.01 | 3.05±0.02 |
| ENERGY | 768 | 8 | 1.81±0.19 | 0.91±0.12 | 0.58±0.05 | **0.56±0.04** | 0.60±0.03 | 2.17±0.02 | 1.13±0.02 | 0.95±0.09 | 0.68±0.02 | 0.54±0.02 |
| KIN8NM | 8192 | 8 | -0.86±0.38 | **-1.2±0.03** | -1.15±0.01 | -1.12±0.01 | -1.05±0.00 | -0.81±0.01 | -0.83±0.01 | -0.92±0.02 | N/A±0.00 | N/A±0.00 |
| POWER | 9568 | 4 | 3.35±0.22 | 2.74±0.02 | **2.69±0.00** | 2.92±0.00 | 2.90±0.00 | 2.83±0.01 | 2.88±0.00 | 2.81±0.00 | N/A±0.00 | N/A±0.00 |
| PROTEIN | 45730 | 9 | **2.84±0.04** | 2.87±0.0 | 2.85±0.00 | 2.87±0.00 | 2.96±0.02 | 3.00±0.00 | 2.99±0.00 | 2.90±0.00 | N/A±0.00 | N/A±0.00 |
| RED WINE | 1588 | 11 | 0.97±0.02 | **0.76±0.08** | 0.97±0.06 | 0.97±0.02 | 1.20±0.04 | 1.01±0.02 | 0.97±0.02 | 1.01±0.02 | 1.04±0.01 | 0.26±0.03 |
| YACHT | 308 | 6 | 2.37±0.55 | 0.29±0.1 | 0.59±0.11 | **-0.02±0.07** | 0.59±0.13 | 1.11±0.04 | 1.22±0.18 | 0.79±0.11 | 1.03±0.03 | 0.10±0.05 |
| NAVAL | 11934 | 16 | **-7.25±0.08** | -6.76±0.1 | -7.21±0.06 | -5.62±0.04 | -4.11±0.00 | -2.80±0.00 | -2.80±0.00 | -2.97±0.14 | -7.13±0.02 | N/A±0.00 |
| Mean Rank | | | 5.5 | **2.06** | 2.22 | 3.33 | 4.94 | 7 | 6.11 | 4.83 | | |

Table 1: The table shows the average test NLL on several UCI regression datasets. We train on random $90\%$ of the data and predict on $10\%$. This is repeated 10 times and we report mean and standard deviation. The results for our competitors are taken from Ma and Hernández-Lobato [2021]. One can see that GWI-net obtains the best mean rank of all methods being the best model on 4/9 datasets and performing competitively on all datasets. Note that we exclude FBNN and exact Gaussian processes from the comparison because their computational complexity is often prohibitively large.

**Classification and OOD Detection** We demonstrate the ability of GWI to perform image classifications on Fashion MNIST [Xiao et al., 2017] and CIFAR-10 [Krizhevsky et al., 2009]. We compare to FVI, mean-field variational inference (MVFI) [Blundell et al., 2015], maximum a posteriori approximation (MAP), K-FAC Laplace-GNN [Martens and Grosse, 2015] and its dampened version [Ritter et al., 2018]. Implementation details are discussed in A.8.

We also assess the ability of our model to perform out-of-distribution detection using in-distribution (ID) / out of-distribution (OOD) pairs given as FashionMNIST/MNIST and CIFAR10/SVNH. Following the setting of Osawa et al. [2019], Immer et al. [2021] we calculate the area under the curve (AUC) of a binary out-of-distribution classifier based on predictive entropies. Results are shown in Table 2.

| **Model** | FMNIST | | | CIFAR 10 | | |
|---|---|---|---|---|---|---|
| | Accuracy | NLL | OOD-AUC | Accuracy | NLL | OOD-AUC |
| GWI-net | **93.25 ±0.09** | **0.250 ±0.00** | **0.959 ±0.01** | **83.82 ±0.00** | **0.553 ±0.00** | 0.618 ±0.00 |
| FVI | 91.60±0.14 | 0.254±0.05 | 0.956±0.06 | 77.69 ±0.64 | 0.675±0.03 | 0.883±0.04 |
| MFVI | 91.20±0.10 | 0.343±0.01 | 0.782±0.02 | 76.40±0.52 | 1.372±0.02 | 0.589±0.01 |
| MAP | 91.39±0.11 | 0.258±0.00 | 0.864±0.00 | 77.41±0.06 | 0.690±0.00 | 0.809±0.01 |
| KFAC-LAPLACE | 84.42±0.12 | 0.942±0.01 | 0.945±0.00 | 72.49±0.20 | 1.274±0.01 | 0.548±0.01 |
| RITTER et al. | 91.20±0.07 | 0.265±0.00 | 0.947±0.00 | 77.38±0.06 | 0.661±0.00 | 0.796±0.00 |

Table 2: We report average accuracy, NLL and OOD-AUC on test data for 10 different train/test splits. The results for FVI are obtained from Ma and Hernández-Lobato [2021] and for MAP, KFAC and Ritter et al. results are taken from Immer et al. [2021] .

Our method performs best in all categories on the Fashion MNIST dataset achieving state-of-the-art results. On CIFAR10 we obtain the highest accuracy and best NLL by a significant margin and perform competitively in the OOD detection task.

## 6 Limitations

In this section we discuss some of the shortcomings and difficulties which are related to our method.

The GVI-FS framework allows the specification of function space inference via infinite dimensional parameters such as mean and kernel functions. This great flexibility essentially allows the specification of mismatched prior and posterior parameters. We illustrate such a case in Appendix A.10.

GWI-net relies on the SVGP kernel defined in 18 for its posterior approximation. It therefore inherits numerical instabilities associated with the inversion of the kernel matrix. For the data sets discussed in this paper it was possible to overcome these issues by smart initialisation of the optimiser (cf. Appendix A.7), but it may be an interesting research avenue to come up with a kernel that avoids these instabilities.

Our method approximates the Wasserstein distance in function space via the spectrum of kernel matrices (cf. Appendix A.4). These approximations require quick spectral decay of the composition

of prior and variational covariance operator to be accurate and computationally tractable. The prior SE kernel combined with the variational SVGP kernel did have this property (cf. A.13) which allowed for cheap and accurate approximations. However, other parameterisations may result in less accurate estimation. A theoretical investigation of how the approximation quality relates to kernel properties is an interesting topic for further research.

The proposed framework models prior and variational distribution with a Gaussian measure on the space of square integrable functions. As a consequence the posterior distribution for the functional output is Gaussian as well. This means it is unimodal and concentrated around the posterior mean. Although this constrains the form of functional posterior significantly the authors would argue that the empirical success of GWI-net demonstrates that the approach is flexible to meaningfully quantify uncertainty.

## 7   Conclusion

In this paper, we developed a framework for generalized variational inference in infinite-dimensional function spaces. We leveraged the function space perspective to develop a new inference approach combining Gaussian measures and Wasserstein distance with predictive performance of deep neural networks, yielding principled uncertainty quantification. The value of our method was demonstrated on several benchmark datasets.

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
