# A  Appendix

## A.1  Bayesian Inference as an Optimization Problem for an Infinite-Dimensional Prior Measure

Let $E$ be a (infinite dimensional) Polish space and $\mathcal{B}(E)$ the Borel $\sigma$-algebra on $E$. We denote the set of Borel probability measures on $\mathcal{B}(E)$ as $\mathcal{P}(E)$ and choose a fixed prior measure $P \in \mathcal{P}(E)$. The likelihood is described by a Markov kernel function $p : \mathcal{Y} \times E \to [0, \infty)$ with

$$(y, f) \mapsto p(y|f), \tag{19}$$

where $\mathcal{Y} \subset \mathbb{R}^N$ is Borel measurable. The prior and the likelihood induce for any fixed $y \in \mathcal{Y}$ a posterior measure denoted as $\widehat{P} \in \mathcal{P}(E)$ [Ghosal and Van der Vaart, 2017, Chapter 1.3].

The next theorem shows that this posterior measure is the solution to a certain optimization problem.

**Theorem 2** (Bayes Posterior as optimization). *The Bayesian posterior measure $\widehat{P}$ is given as*

$$\widehat{P} = \underset{Q \in \mathcal{P}(E)}{argmin} \ \left\{ -\mathbb{E}_Q \big[ \log p(y|F) \big] + \mathbb{D}_{KL}(Q, P) \right\} \tag{20}$$

*for any fixed prior measure $P \in \mathcal{P}(E)$ and fixed $y \in \mathcal{Y}$ such that $f \in E \mapsto p(y|f) > 0$.*

*Proof.* According to Bayes rule in infinite dimensions [Ghosal and Van der Vaart, 2017, Chapter 1.3] we know that $\widehat{P}$ is dominated by $P$ with Radon-Nikodym derivative given as

$$\frac{d\widehat{P}}{dP}(f) = \frac{p(y|f)}{p(y)}, \tag{21}$$

for $f \in E$ where $p(y) := \int p(y|F = f) \, dP(f)$ is the marginal likelihood for $y$. The reverse is also true and $P$ is dominated by $\widehat{P}$. We prove this by contraposition and therefore assume that $P(A) > 0$ for some $A \in \mathcal{B}(E)$. From Bayes rule we know that

$$\widehat{P}(A) = \int_A \frac{p(y|f)}{p(y)} \, dP(f) > 0 \tag{22}$$

as the integrand is positive by assumption and $P(A) > 0$. This gives $\widehat{P}(A) > 0$ and therefore that $P$ is dominated by $\widehat{P}$. In this case standard rules for Radon-Nikodym derivatives give that

$$\frac{dP}{d\widehat{P}}(f) = \frac{p(y)}{p(y|f)}, \tag{23}$$

for $f \in E$. Note that without loss of generality we can assume that the optimal $Q \in \mathcal{P}(E)$ is dominated by $P$ (and therefore also dominated by $\widehat{P}$) since otherwise (20) is infinite by definition of the KL divergence. For such a $Q$ dominated by $P$ it holds that

$$L(Q) := -\mathbb{E}_Q \big[ \log p(y|F) \big] + \mathbb{D}_{KL}(Q, P) \tag{24}$$

$$= -\int \log p(y|f) \, dQ(f) + \int \log \frac{dQ}{dP}(f) \, dQ(f) \tag{25}$$

$$= -\int \log p(y|f) \, dQ(f) + \int \log \frac{dQ}{d\widehat{P}}(f) \, dQ(f) + \int \log \frac{d\widehat{P}}{dP}(f) \, dQ(f), \tag{26}$$

where the last line follows from the chain rule for Radon-Nikodym derivatives. We further have

$$L(Q) = -\int p(y|f) \, dQ(f) + \mathbb{D}_{KL}(Q, \widehat{P}) + \int \frac{p(y|f)}{p(y)} \, dQ(f) \quad \text{(Bayes Rule)} \tag{27}$$

$$= \mathbb{D}_{KL}(Q, \widehat{P}) + p(y) \tag{28}$$

$$\geq p(y), \tag{29}$$

since $\mathbb{D}_{KL}(Q, P) \geq 0$, with equality if and only if $Q = \widehat{P}$. This proves the claim. $\qquad \square$

## A.2 Pointwise Evaluation as Weak Limit

To outline the problem briefly: If $F \sim \mathcal{N}(m, C)$ is a GRE with mean $m \in L^2(\mathcal{X}, \rho, \mathbb{R})$ and covariance operator $C$ as defined in (9) then it is in general unclear what the distribution of $F(x)$ would be for a fixed $x \in \mathcal{X}$. The technical reason is that the pointwise evaluation $\pi_x : L^2(\mathcal{X}, \rho, \mathbb{R}) \to \mathbb{R}$, i.e.

$$\pi_x(f) := f(x) \tag{30}$$

is not well-defined. An element $g$ of the space $L^2(\mathcal{X}, \rho, \mathbb{R})$ is an equivalence class and only identifiable up to a $\rho$-nullset. This means that the definition of $\pi_x$ in (30) makes no sense whenever $\rho(\{x\}) = 0$ which is the case whenever $\rho$ has a pdf w.r.t. the Lebesgue measure.

However, we will remedy this situation by defining for a fixed $x \in \mathcal{X}$

$$F(x) := \lim_{n \to \infty} \langle F, h_{n,x} \rangle_2 \tag{31}$$

where $h_{n,x} \in L^2(\mathcal{X}, \rho, \mathbb{R})$ is an appropriately chosen sequence and the limit is to be understood as convergence in distribution of the sequence of scalar random variables $\langle F, h_{n,x} \rangle_2$.

**Theorem 3.** *Let $F \sim \mathcal{N}(m, C)$ be a GRE in $\mathcal{L}^2(\mathcal{X}, \rho, \mathbb{R})$ with mean $m \in L^2(\mathcal{X}, \rho, \mathbb{R})$ and covariance operator $C$ as defined in (9). Assume that $\rho$ is a probability measure on $\mathcal{X} \subset \mathbb{R}^D$ and that $\rho$ is absolutely continuous with respect to the Lebesgue measure $\lambda$ on $\mathbb{R}^D$ with pdf $\rho'$. Denote the support of the measure $\rho$ by $supp(\rho)$ and assume that $x$ is an arbitrary point in the interior of $supp(\rho)$ such that $m$, $k$ and $\rho'$ are continuous at $x$.*

*Let*

$$\eta(t) = \begin{cases} \exp\left(-\frac{1}{1-|t|^2}\right) & \text{if } |t| < 1, \\ 0 & \text{if } |t| \geq 1. \end{cases} \tag{32}$$

*be the so called standard molifier and note that $\eta$ is smooth with $\int \eta(t) \, dt = 1$. We further define the sequence $h_{n,x}(t) := \eta\big(n(t-x)\big)/\rho'(t)$ for $n \in \mathbb{N}$, $t \in supp(\rho)$ and $h_{n,x} = 0$ for $t \notin supp(\rho)$. Then*

$$\langle F, h_{n,x} \rangle_2 \xrightarrow{\mathcal{D}} \mathcal{N}\big(m(x), k(x,x)\big) \tag{33}$$

*for $n \to \infty$ where $\xrightarrow{\mathcal{D}}$ denotes convergence in distribution.*

*Proof.* Note that $supp(h_{n,x}) = B_{1/n}(x) := \{t \in \mathbb{R}^D : |t - x| \leq \frac{1}{n}\}$ and $B_{1/n}(x) \subset supp(\rho)$ for large enough $n \in \mathbb{N}$ since $x$ is from the interior of $supp(\rho)$. This means that $h_{n,x} \in L^2(\mathcal{X}, \rho, \mathbb{R})$ for large enough $n$ as

$$\int h_{n,x}(t) \, d\rho(t) = \int_{supp(\rho)} \left(\frac{\eta\big(n(t-x)\big)}{\rho'(t)}\right)^2 \rho'(t) \, d\lambda(t) \tag{34}$$

$$= \int_{supp(\rho)} \frac{\eta\big(n(t-x)\big)}{\rho'(t)} \, dt \tag{35}$$

$$= \int_{B_{1/n}(x)} \frac{\eta\big(n(t-x)\big)}{\rho'(t)} \, dt. \tag{36}$$

The last expression is finite for large enough $n$ because the integrand is continuous at $x$. According to the definition of of GREs we therefore conclude that

$$\langle F, h_{n,x} \rangle_2 \sim \mathcal{N}\big(\langle m, h_{n,x} \rangle_2, \langle C h_{n,x}, h_{n,x} \rangle_2\big) \tag{37}$$

for large enough $n \in \mathbb{N}$.

The next statement we show is that $m_n(x) := \langle m, h_{n,x} \rangle_2 \to m(x)$ for $n \to \infty$. To this end notice that

$$|m_n(x) - m(x)| = |\int_{B_{1/n}(x)} h_{n,x}(t)\big(m(x) - m(t)\big) \, d\rho(t)| \tag{38}$$

$$\leq \int_{B_{1/n}(x)} \eta\Big(n(t-x)\Big)|m(x) - m(t)| \, dt. \tag{39}$$

Let now $\epsilon > 0$ be arbitrary. For $n$ large enough we $|m(x) - m(t)| \leq \epsilon$ for all $t \in B_{1/n}(x)$ due to the continuity of $m$ in $x$. This immediately implies

$$\int_{B_{1/n}(x)} \eta\Big(n(t-x)\Big)|m(x) - m(t)| \, dt \leq \epsilon \int_{B_{1/n}(x)} \eta\Big(n(t-x)\Big) \, dt = \epsilon, \tag{40}$$

for large enough $n$ which shows the convergence of $m_n(x)$ to $m(x)$.

A similar argument shows that $k_n(x,x) := \langle Ch_{n,x}, h_{n,x}\rangle_2 \to k(x,x)$ for $n \to \infty$.

We therefore conclude that

$$\langle F, h_{n,x}\rangle_2 = \langle F, h_{n,x}\rangle_2 - m_n(x) + m_n(x) \tag{41}$$

$$= \sqrt{k_n(x,x)} \underbrace{\frac{\langle F, h_{n,x}\rangle_2 - m_n(x)}{\sqrt{k_n(x,x)}}}_{\sim \mathcal{N}(0,1)} + m_n(x) \tag{42}$$

$$\xrightarrow{\mathcal{D}} \mathcal{N}\big(m(x), k(x,x)\big) \tag{43}$$

for $n \to \infty$ due to Slutsky's theorem. $\qquad \square$

According to Theorem 3 we can simply define $F(x) \sim \mathcal{N}(m(x), k(x,x))$ for all $x$ in the interior of the support of $\rho$ if $m$, $k$ and $\rho'$ are continuous at $x$. These are mild assumptions and we can typically assume that they are satisfied in practice.

## A.3 The Wasserstein Metric for Probability Measures

Let $E$ be a Polish space. For $p \geq 1$, let $P_p(E)$ denote the collection of all probability measures $\mu$ on $E$ with finite $p^{\text{th}}$ moment, that is, there exists some $x_0$ in $M$ such that:

$$\int_M d(x, x_0)^p \, d\mu(x) < \infty. \tag{44}$$

The $p^{\text{th}}$ Wasserstein distance between two probability measures $\mu$ and $\nu$ in $P_p(E)$ is defined as

$$W_p(\mu, \nu) := \left( \inf_{\gamma \in \Gamma(\mu,\nu)} \int_{E \times E} d(x,y)^p \, d\gamma(x,y) \right)^{1/p}, \tag{45}$$

where $\Gamma(\mu, \nu)$ denotes the collection of all measures on $E \times E$ with marginals $\mu$ and $\nu$ on the first and second arguments respectively.

More details about the Wasserstein distance can be found in Chapter 7 of Ambrosio et al. [2005].

## A.4 A Tractable Approximation of the Wasserstein Metric

Recall that the Wasserstein metric for the two Gaussian measures $P = \mathcal{N}(m_P, C_P)$ and $Q = \mathcal{N}(m_Q, C_Q)$ on the Hilbert space $H = L^2(\mathcal{X}, \rho, \mathbb{R})$ is given as

$$W_2^2(P,Q) = \|m_P - m_Q\|_2^2 + tr(C_P) + tr(C_Q) - 2 \cdot tr\Big[(C_P^{1/2} C_Q C_P^{1/2})^{1/2}\Big]. \tag{46}$$

Further the operators $C_P$ and $C_Q$ are defined through trace-class kernels $k$ and $r$ as described in Section 3.1. We will now discuss how to approximate each term in (46).

First, note that

$$\|m_P - m_Q\|_2^2 = \int \big(m_P(x) - m_Q(x)\big)^2 \, d\rho(x) \approx \frac{1}{N} \sum_{n=1}^{N} \big(m_P(x_n) - m_Q(x_n)\big)^2, \tag{47}$$

which follows by replacing the true input distribution with the empirical data distribution. Second, note that under very general conditions on $k$ and $\rho$ it holds that [Brislawn, 1991]

$$tr(C_P) = \int k(x,x)\,d\rho(x) \tag{48}$$

and similarly for $C_Q$. Again by replacing $\rho$ with the empirical data distribution we obtain natural estimators:

$$tr(C_P) \approx \frac{1}{N} \sum_{n=1}^{N} k(x_n, x_n), \tag{49}$$

$$tr(C_Q) \approx \frac{1}{N} \sum_{n=1}^{N} r(x_n, x_n). \tag{50}$$

Denote by $\lambda_n(C)$ the $n$-th eigenvalue of a positive, self-adjoint operator $C$. By definition of the trace and the square root of an operator we have

$$tr\left[(C_P^{1/2} C_Q C_P^{1/2})^{1/2}\right] = \sum_{n=1}^{\infty} \sqrt{\lambda_n\left(C_P^{1/2} C_Q C_P^{1/2}\right)} \tag{51}$$

$$= \sum_{n=1}^{\infty} \sqrt{\lambda_n\left(C_Q C_P\right)}, \tag{52}$$

where the second line follows from the fact that the operator $C_Q C_P$ has the same eigenvalues as $C_P^{1/2} C_Q C_P^{1/2}$ [Hladnik and Omladič, 1988, Proposition 1]. The operator $C_Q C_P$ is given as

$$C_Q C_P g(x) = \int r(x,x')(C_P f)(x')\,d\rho(x') \tag{53}$$

$$= \int r(x,x')\left(\int k(x',t)f(t)d\rho(t)\right)d\rho(x') \tag{54}$$

$$= \int\int r(x,x')k(x',t)f(t)\,d\rho(x')d\rho(t) \tag{55}$$

$$= \int (r*k)(x,t)f(t)\,d\rho(t), \tag{56}$$

where we define

$$(r*k)(x,t) := \int r(x,x')k(x',t)\,d\rho(x') \tag{57}$$

for all $x,t \in \mathcal{X}$. This means that $C_Q C_P$ is also an integral operator with (non-symmetric) kernel $r*k$. We again replace $\rho$ with $\widehat{\rho}$ to obtain

$$\widehat{(r*k)}(x,t) = \frac{1}{N} \sum_{n=1}^{N} r(x,x_n)k(x_n,t). \tag{58}$$

The spectrum of $C_Q C_P$ can now be approximated by the spectrum of the matrix $\frac{1}{N}\widehat{(r*k)}(X,X)$ [Rasmussen, 2003, cf. Chapter 4.3.2] or $\frac{1}{N_S}\widehat{(r*k)}(X_S,X_S)$ where $X_S$ is a subsample of the data points $X$ of size $N_S < N$. If we plug this approximation into (52) we obtain

$$tr\left[(C_P^{1/2} C_Q C_P^{1/2})^{1/2}\right] \approx \sum_{m=1}^{N_S} \sqrt{\lambda_m\left(\frac{1}{N_S}\widehat{(r*k)}(X_S,X_S)\right)} \tag{59}$$

$$= \frac{1}{\sqrt{N_S}} \sum_{m=1}^{N_S} \sqrt{\lambda_m\left(\frac{1}{N}r(X_S,X)k(X,X_S)\right)}, \tag{60}$$

which is the last expression that we had to approximate.

Note that since $C_Q C_P$ has the same spectrum as the self-adjoint, positive, trace-class operator $C_P^{1/2} C_Q C_P^{1/2}$ we know that its eigenvalues are real, positive and converge to zero.

## A.5 Generalized Loss for Regression in Batch Mode

The batch version of the generalized loss is given as:

$$\widehat{\mathcal{L}} = \frac{N}{2}\log(2\pi\sigma^2) + \frac{N}{N_B}\sum_{b=1}^{N_B}\frac{\big(y_{n_b} - m_Q(x_{n_b})\big)^2 + r(x_{n_b}, x_{n_b})}{2\sigma^2} + \frac{1}{N_B}\sum_{b=1}^{N_B}\big(m_P(x_{n_b}) - m_Q(x_{n_b})\big)^2 \tag{61}$$

$$+ \frac{1}{N_B}\sum_{b=1}^{N_B}k(x_{n_b}, x_{n_b}) + \frac{1}{N_B}\sum_{b=1}^{N_B}r(x_{n_b}, x_{n_b}) - \frac{2}{\sqrt{N_B N_S}}\sum_{s=1}^{N_S}\sqrt{\lambda_s\big(r(X_S, X_B)k(X_B, X_S)\big)}, \tag{62}$$

$N_B \in \mathbb{N}$ is the batch-size. The indices $n_1, \ldots, n_{N_B}$ are the batch-indices and $X_B$ is the batch matrix.

## A.6 GWI for (Multiclass) Classification

Let $\{(x_n, y_n)\}_{n=1}^N \subset \mathcal{X} \times \mathcal{Y}$ be data with $\mathcal{X} \subset \mathbb{R}^D$ and $\mathcal{Y} = \{1, \ldots, J\}$, where $J \in \mathbb{N}$ represents $J \geq 2$ distinct classes.

**Model** We use the same likelihood for $y := (y_1, \ldots, y_N)$ as described in Chapter 4 of Matthews [2017] which is:

$$p(y|f_1, \ldots, f_J) = \prod_{n=1}^N p(y_n|f_1, \ldots, f_J) \tag{63}$$

with

$$p(y_n|f_1, \ldots, f_J) := h_{y_n}^\epsilon\big(f_1(x_n), \ldots, f_J(x_n)\big), \tag{64}$$

for $y_n \in \{1, \ldots, J\}$. The function $h_\ell^\epsilon$ is defined as

$$h_\ell^\epsilon(t_1, \ldots, t_J) \begin{cases} 1 - \epsilon & \text{if } \ell = \underset{j=1,\ldots,J}{\mathrm{argmax}}\{t_j\}, \\ \frac{\epsilon}{J-1} & \text{if otherwise.} \end{cases} \tag{65}$$

for $\ell = 1, \ldots, J$ for $\epsilon > 0$. We chose $\epsilon = 1\%$ in our implementation.

We assume that $F_1, \ldots F_J$ are independent GREs on $L^2(\mathcal{X}, \rho, \mathbb{R})$ with prior means $m_{P,j}$ and prior covariance operators $C_{P,j}$, $j = 1, \ldots, J$.

The variational measures for $F_1, \ldots, F_J$ are assumed to be independent and given as $Q_j = \mathcal{N}(m_{Q,j}, C_{Q,j})$ for $j = 1, \ldots, J$. We further write $\mathbb{Q}\big(\big(F_1(x), \ldots, F_J(x)\big) \in A\big)$, $A \subset \mathbb{R}^J$ for the variational (posterior) approximation of the probability of the event $\{\big(F_1(x), \ldots, F_J(x)\big) \in A\}$.

This leads to the following expected log-likelihood

$$\mathbb{E}_{\mathbb{Q}}\big[\log p(y|F_1, \ldots, F_J)\big] \tag{66}$$

$$= \sum_{n=1}^N \mathbb{E}_{\mathbb{Q}}\big[\log p(y_n|F_1, \ldots, F_J)\big] \tag{67}$$

$$= \sum_{n=1}^N \log(1-\epsilon)\mathbb{Q}\big(\underset{j=1,\ldots,J}{\mathrm{argmax}}\{F_j(x_n)\} = y_n\big) + \log\big(\frac{\epsilon}{J-1}\big)\mathbb{Q}\big(\underset{j=1,\ldots,J}{\mathrm{argmax}}\{F_j(x_n)\} \neq y_n\big) \tag{68}$$

$$\approx \sum_{n=1}^N \log(1-\epsilon)S(x_n, y_n) + \log\big(\frac{\epsilon}{J-1}\big)\big(1 - S(x_n, y_n)\big), \tag{69}$$

with

$$S(x, j) := \frac{1}{\sqrt{\pi}}\sum_{i=1}^I w_i \prod_{l \neq j}\phi\Big(\frac{\sqrt{2r_j(x,x)}\xi_i + m_{Q,j}(x) - m_{Q,l}(x)}{\sqrt{r_l(x,x)}}\Big) \tag{70}$$

for any $x \in \mathcal{X}$, $j = 1, \ldots, J$ where $(w_i, \xi_i)_{i=1}^I$ are the weights and roots of the Hermite polynomial of order $I \in \mathbb{N}$. This is the same Gauss-Hermite approximation as described in Chapter 4 of Matthews [2017].

The final objective for multiclass classification is given as

$$\mathcal{L} = -\mathbb{E}_Q\big[\log p(y|F_1, \ldots, F_J)\big] + \sum_{j=1}^{J} W_2^2(P_j, Q_j), \tag{71}$$

where the expected log-likelihood is approximated by (69) and each Wasserstein distance $W_2^2(P_j, Q_j)$ can be estimated as in (14)-(15).

**Prediction**     The probability that an unseen point $x^* \in \mathcal{X}$ belongs to class $j \in \{1, \ldots, J\}$ is given as

$$\mathbb{Q}(Y^* = j) = (1 - \epsilon)S(x^*, j) + \frac{\epsilon}{J - 1}\big(1 - S(x^*, j)\big) \tag{72}$$

for any $x^* \in \mathcal{X}$. We predict the class label as maximiser of this probability. If we apply tempering, we simply replace every $r_j(x, x)$ with $T \cdot r_j(x, x)$ for $j = 1, \ldots, J$ in the definition of $S(x, j)$.

**Negative Log Likelihood**     The variational approximation to the negative log-likelihood is

$$NLL = -\log\Big[(1 - \epsilon)S(x^*, y^*) + \frac{\epsilon}{J - 1}\big(1 - S(x^*, y^*)\big)\Big] \tag{73}$$

for any point $x^* \in \mathcal{X}$ for which we know that the class label is $y^* \in \{1, \ldots, J\}$.

### A.7    Implementation Details: Regression

The Regression model is given as $F \sim \mathcal{N}(0, C)$ and

$$Y_n = F(x_n) + \epsilon_n \tag{74}$$

with $\epsilon_n \sim \mathcal{N}(0, \sigma^2)$, $n = 1, \ldots, N$. The covariance operator $C_P$ depends on the choice of a kernel $k$, i.e. $C_P = C_{P,k}$ for which we use the ARD kernel $k$ given as

$$k(x, x') = \sigma_f^2 \exp\Big(-\frac{1}{2}\sum_{d=1}^{D} \frac{(x_d - x_d')^2}{\alpha_d^2}\Big) \tag{75}$$

for $x, x' \in \mathbb{R}^D$. We refer to $\sigma_f > 0$ as *kernel scaling factor*, to $\alpha_d > 0$ as *length-scale* for dimension $d$ and to $\sigma > 0$ as *observation noise*.

The data is first randomly split into three categories: training set $80\%$, validation set $10\%$ and test set $10\%$. The observations $Y$ are then standardised by subtracting the empirical mean (of the training data) and dividing by the empirical standard deviation (of the training data). The inputs data $X$ is left unaltered.

**The number of inducing points**     The number of inducing points $M$ is treated as a hyperparameter, this means we train the model for each $M \in \{0.5\sqrt{N}, \sqrt{N}, 1.5\sqrt{N}, 2\sqrt{N}\}$ and choose the best model. For GWI: SVGP we use $M \in \{1\sqrt{N}, 2\sqrt{N}, \ldots 5\sqrt{N}\}$.

**The choice of inducing points**     The input points $Z_1, \ldots, Z_M$ in (18) are sampled independently from the training data $X$ and then fixed for GWI-net. For GWI: SVGP they are only initialised this way and then learned by maximising the generalized loss.

**Prior hyperparameters**     The prior hyperparameters $\sigma_f$, $\alpha := (\alpha_1, \ldots, \alpha_D)$ and $\sigma$ are chosen by maximising the marginal log-likelihood for the data $X = Z$ and the corresponding observations, which we denote $Y_Z$. Note that the marginal log-likelihood is tractable and given as

$$\log p(y_Z) = -\frac{1}{2}\log\Big(\det\big(k(Z, Z) + \sigma^2 I_M\big)\Big) - \frac{1}{2}y_Z^T\big(k(Z, Z) + \sigma^2 I_M\big)^{-1}y_Z. \tag{76}$$

and can therefore be evaluated in $\mathcal{O}(M^3) = \mathcal{O}(N\sqrt{N})$. **Variational mean**     For GWI-net we use a neural network with $L = 2$ hidden layers, width $D_1 = D_2 = 10$ and tanh as activation function. This follows the set-up of Ma and Hernández-Lobato [2021].

**Variational kernel**     The kernel $r$ which is chosen as described in (18) and therefore depends on the covariance matrix $\Sigma \in \mathbb{R}^{M \times M}$ and the $M \in \mathbb{N}$ inducing points $Z = (Z_1, \ldots, Z_M) \in \mathbb{R}^{D \times M}$. We parametrise $\Sigma$ as $\Sigma = LL^T$ with initialisation

$$L = \text{Chol}\Big(\big(k(Z, Z) + \frac{1}{\sigma^2}k(Z, X)k(X, Z)\big)^{-1}\Big), \tag{77}$$

where $k(Z, X)k(X, Z)$ is approximated by batch-sizing as $\frac{N}{N_B}k(Z, X_B)k(X_B, Z)$. This corresponds to an approximation of the optimal choice for $\Sigma$ in SVGP [Titsias, 2009].

**Parameters in the generalized loss**     The generalized loss in Appendix A.5 depends further on $N_S$, $N_B$ and $X_S$. The batch-size $N_B$ is chosen to be $N_B = 1000$ for $N > 1000$. For $N < 1000$ we use the full training data. The comparison points $X_S$ are sampled independently from the training data $X$ in each iteration. We train here for 1000 epochs on the regression task and 100 epochs on the classification task following Ma and Hernández-Lobato [2021].

### Tempering the predictive posterior

Wenzel et al. [2020] observe that the performance of many Bayesian neural networks can be improved by *tempering* the predictive posterior. Tempering refers to a shrinking of the predictive posterior variance by a factor of $\alpha_T \in [0, 1]$. This effect has also been observed for Gaussian processes in Adlam et al. [2020] where it can be interpreted as elevating problems that occur from prior misspecification. The prior hyperparameters for the ARD kernel $k$ in (16) are selected by maximising the marginal log-likelihood on a subset of the training data. This procedure may lead to prior misspecification, which is why we decided to temper the predictive posterior, which means that we use the predictive distribution

$$Y^*|Y \sim \mathcal{N}\Big(m_Q(x^*), \alpha_T\big(r(x^*, x^*) + \sigma^2\big)\Big) \tag{78}$$

for an unseen data point $x^* \in \mathcal{X}$. The (tempered) NLL for each data point is given as

$$\text{NLL} := -\log p_{\alpha_T}(y^*|y) \tag{79}$$

$$= \frac{1}{2}\log\Big(\alpha_T \cdot (r(x^*, x^*) + \sigma^2)\Big) + \frac{1}{2}\frac{(y - y^*)^2}{\alpha_T \cdot (r(x^*, x^*) + \sigma^2)} + \frac{1}{2}\log(2\pi). \tag{80}$$

The tempering factor $\alpha_T$ is chosen as minimiser of the average NLL on the validation set. The final predictions on the test set are made using this optimal $\alpha_T$ and (78). Note however that for the NLL numbers reported in Table 1 we add $\log(\widehat{\sigma}_{train})$ to (80) where $\widehat{\sigma}_{train}$ is the empirical standard deviation of the training data. This is done for fair comparison as it is how the NLL is calculated in Ma and Hernández-Lobato [2021].

## A.8   Implementation Details: Classification

As described in section (A.6) we use the prior mean functions $m_{P,j}$ and kernels $k_j$ for $j = 1, \ldots, J$. For our experiments we chose $m_{P,j} = 0$ for $j = 1, \ldots, J$ and $k := k_1 = \ldots, k_J$ where $k$ is the ARD kernel in (16).

We use a multi-output neural network for the variational means $m_{Q,j}$ and an SVGP kernel for each $r_j$, $j = 1, \ldots, J$.

**The number of inducing points**     The number of inducing points $M$ is treated as a hyperparameter, this means we train the model for each $M \in \{0.5\sqrt{N}, 0.75\sqrt{N}, \sqrt{N}\}$ and choose the best model.

**The choice of inducing points**     The input points $Z_1, \ldots, Z_M$ in (18) are sampled independently from the training data $X$ and then fixed for GWI-net.

**Prior hyperparameters**     The prior hyperparameters are initialised as described in A.7, thus maximising the marginal likelihood of a *regression* model, since the marginal likelihood of our classification model is intractable.

**Variational mean**     We use the same CNN architecture as described in Immer et al. [2021], Schneider et al. [2019] for all models.

**Variational kernel**     Each variational kernel $r_j$ uses the same inducing points $Z$ but gets an individual matrix $\Sigma^j \in \mathbb{R}^{M \times M}$ for $j = 1, \ldots, J$. They are all initialised as described in A.7.

**Parameters in the generalized loss**     The generalized loss in Appendix A.5 depends on $N_S$, $N_B$ and $X_S$. The batch-size $N_B$ is chosen to be $N_B = 1000$ for $N > 1000$. For $N < 1000$ we use the full training data. The comparison points $X_S$ are sampled independently from the training data $X$ in each iteration. We train 100 epochs on the classification task following Ma and Hernández-Lobato [2021].

**Tempering the predictive posterior** For the same reasons as outlined in Appendix A.7 we temper the predictive posterior. Recall that the NLL for classification is given as

$$NLL = -\log\left[(1-\epsilon)S(x^*, y^*) + \frac{\epsilon}{J-1}\big(1 - S(x^*, y^*)\big)\right] \quad (81)$$

for any point $x^* \in \mathcal{X}$ for which we know that the class label is $y^* \in \{1, \dots, J\}$. We use a tempering factor $\alpha_j > 0$ for each variational measure $Q_j \sim \mathcal{N}(m_{Q,j}, \alpha_j r_j)$, $j = 1, \dots, J$. We train the model with $\alpha_j = 1$ for all $j = 1, \dots, J$ and select the tempering factors afterwards as minimiser of the average NLL on the validation set.

## A.9 Illustrative Example for Two Dimensional Inputs

In Foong et al. [2020] it is observed that several BNN posterior approximation techniques struggle with the quantification of in-between uncertainty. The red points mark where observations were made and it is clear that mean-field variational inference (MVFI) [Hinton and Van Camp, 1993] and Monte Carlo Dropout (MCDO) [Gal and Ghahramani, 2016] exhibit unjustifiably high posterior certainty in the area where no observations are made. This is a pathology of the approximation technique as the true Bayesian posterior which is approximated to very high precision by Hamiltonian Monte Carlo (HMC) [Neal, 2012] or the infinite-width GP limit [Matthews et al., 2018] do not display such behaviour.

In Figure 2 our method GWI-net is displayed next to the methods described in Foong et al. [2020]. As one can observe our model is keenly aware of its limited ability to predict points in-between the two clusters of observed data points.

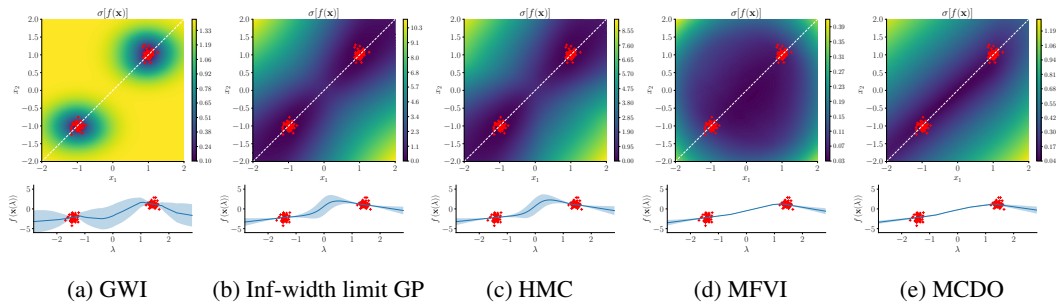

(a) GWI     (b) Inf-width limit GP     (c) HMC     (d) MFVI     (e) MCDO

Figure 2: Regression on a 2D synthetic dataset (red crosses). The colour plots show the standard deviation of the output, $\sigma[f(\mathbf{x})]$, in 2D input space. The plots beneath show the mean with 2-standard deviation bars along the dashed white line (parameterised by $\lambda$). MFVI and MCDO are overconfident for $\lambda \in [-1, 1]$.

## A.10 Model Misspecification in Gaussian Wasserstein Inference

The generalized loss in Appendix A.5 is a valid optimization target for any $m_P, m_Q \in L^2(\mathcal{X}, \rho, \mathbb{R})$ and any trace-class kernels $k$ and $r$. This gives the user a lot of abilities to specify different models, by experimenting with various choices, specifically for $m_Q$ and $r$. However with great power comes great responsibility: it is quite easy to misspecify GWI. To illustrate the issue let us use a periodic kernel $k$ [Duvenaud, 2014] given as

$$k(x, x') := \sigma_f^2 \exp\big(-\frac{1}{\alpha^2}\sin^2(\pi|x - x'|/p)\big) \quad (82)$$

and the SVGP kernel $r$ in (18). By the definition of $r$ the uncertainty will be low for points *similiar* to the inducing points $Z$, i.e. for points $x \in \mathcal{X}$ $k(x, z_m) \approx \sigma_f^2$ for all $m = 1, \dots, M$. A problem now occurs, if the posterior mean $m_Q$ does not respect the knowledge embedded in $k$ and $r$. Lets for example use a simple fully connected deep neural network $m_Q$ and choose the point $x^* := z_1 + 10p$. Assume further that $z_1, \dots, z_M < x^*$. Then we get $k(x^*, z_m) = k(z_1, z_m)$ for all $m = 1, \dots, M$ due to the periodicity of $\sin(x)$ and therefore $r(x^*, x^*) = r(z_1, z_1)$. It is however very unlikely that the neural network will predict $m_Q(z_1)$ as well as $m_Q(x^*)$ since it is unaware of this periodicity.

This small example should illustrate that it is crucial that $m_Q$ is compatible with the prior knowledge reflected in $k$ and $r$. However, note that this problem is not present for our model, GWI-net. The

ARD kernel encodes the inductive bias that the underlying function is infinitely differentiable and that points close to each other have highly correlated functional outputs. A simple fully connected DNN with tanh activation function is indeed smooth and further it is reasonable to assume that predictions are more unreliable the further they are from the data (as measured by the squared euclidean distance). The ARD kernel is in this sense compatible with a fully connected DNN.

It shall be noted that the DNN used for the classification examples in (5) used convolutional layers as explained in Appendix A.8. This can be understood as embedding prior knowledge about translation equivariance into the DNN [Goodfellow et al., 2016, Chapter 9.4]. It might therefore be desirable to use a prior kernel $k$ that embeds similar properties such as the kernel suggested by Van der Wilk et al. [2017]. We considered this to be beyond the scope of this paper but the interaction of DNN architecture and the choice of prior kernels is an interesting avenue for future research.

### A.11 Details on computational resources used

For all our experiments, we distributed our jobs across 8 Nvidia V100 cards.

### A.12 Additional plots for 1D experiments

In Figure 3 we compare GWI-net, GWI-SVGP and SVGP on one-dimensional toy data. Note that all three methods use the same posterior kernel, but GWI-net differs from GWI-SVGP in terms of the posterior mean function. GWI-SVGP and SVGP have the same posterior mean but differ in terms of the objective function used for training.

### A.13 Empirical estimation error of 2-Wasserstein distance

The approximation quality of the 2-Wasserstein distance is determined by the approximation quality of the spectrum of the appearing covariance operators. For most kernels in practice like SE or Matern kernel, the spectrum decays very quickly, which is why using the first 100 eigenvalues often empirically seems to be sufficient to approximate the spectrum and therefore the 2-Wasserstein distance. We plot the magnitude of the first 100 positive eigenvalues (sorted on magnitude) for datasets BOSTON, CONCRETE, ENERGY, WINE and YACHT in Figure 4.

We see in Figure 4 that eigenvalues indeed decay fast.

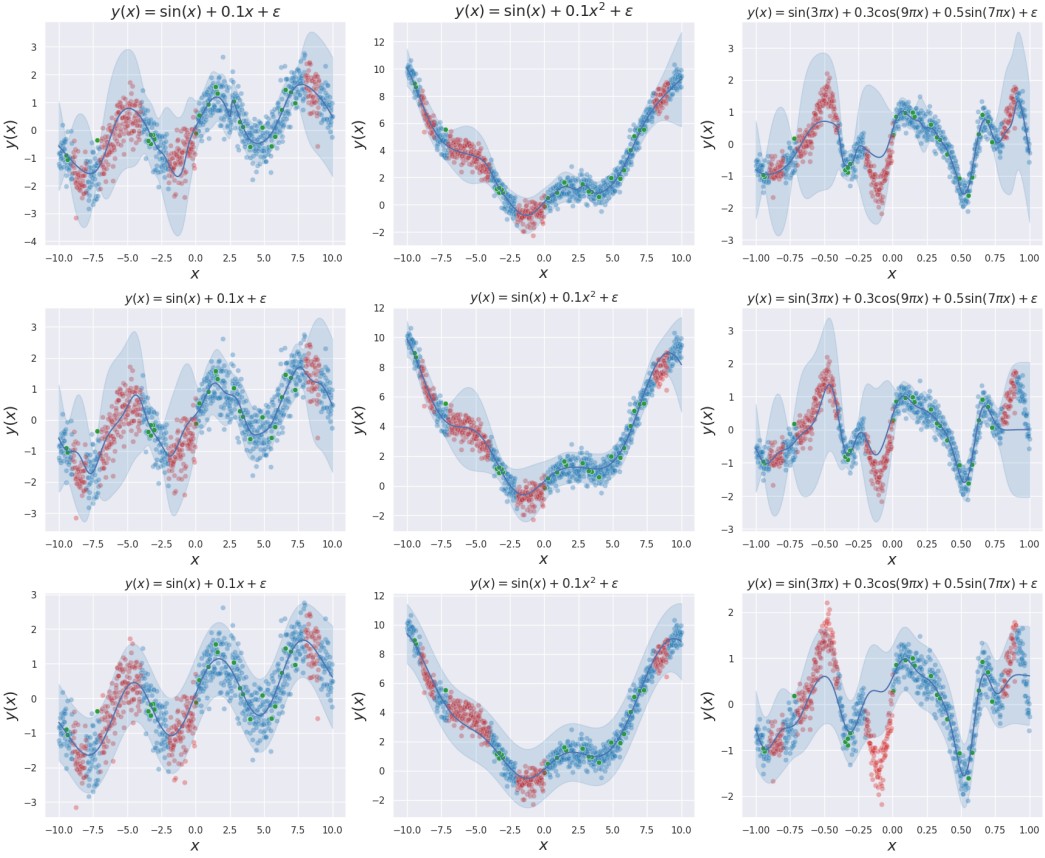

Figure 3: 🟦 : Training data     🟥 : Unseen data     🟩 : Inducing points
We query the above functions at $N = 1000$ equidistant points and add white noise with $\epsilon \sim \mathcal{N}(0, 0.5^2)$. We use $M = 30$ inducing points and train our method as described in Appendix A.7. The plot shows $m_Q(x) \pm 1.96\sqrt{\mathbb{V}[Y^*(x)|Y]}$ where $\mathbb{V}[Y^*(x)|Y]$ is the posterior predictive variance given as $r(x,x) + \sigma^2$. Here the fitted models from top to bottom are GWI-net, GWI-SVGP and SVGP.

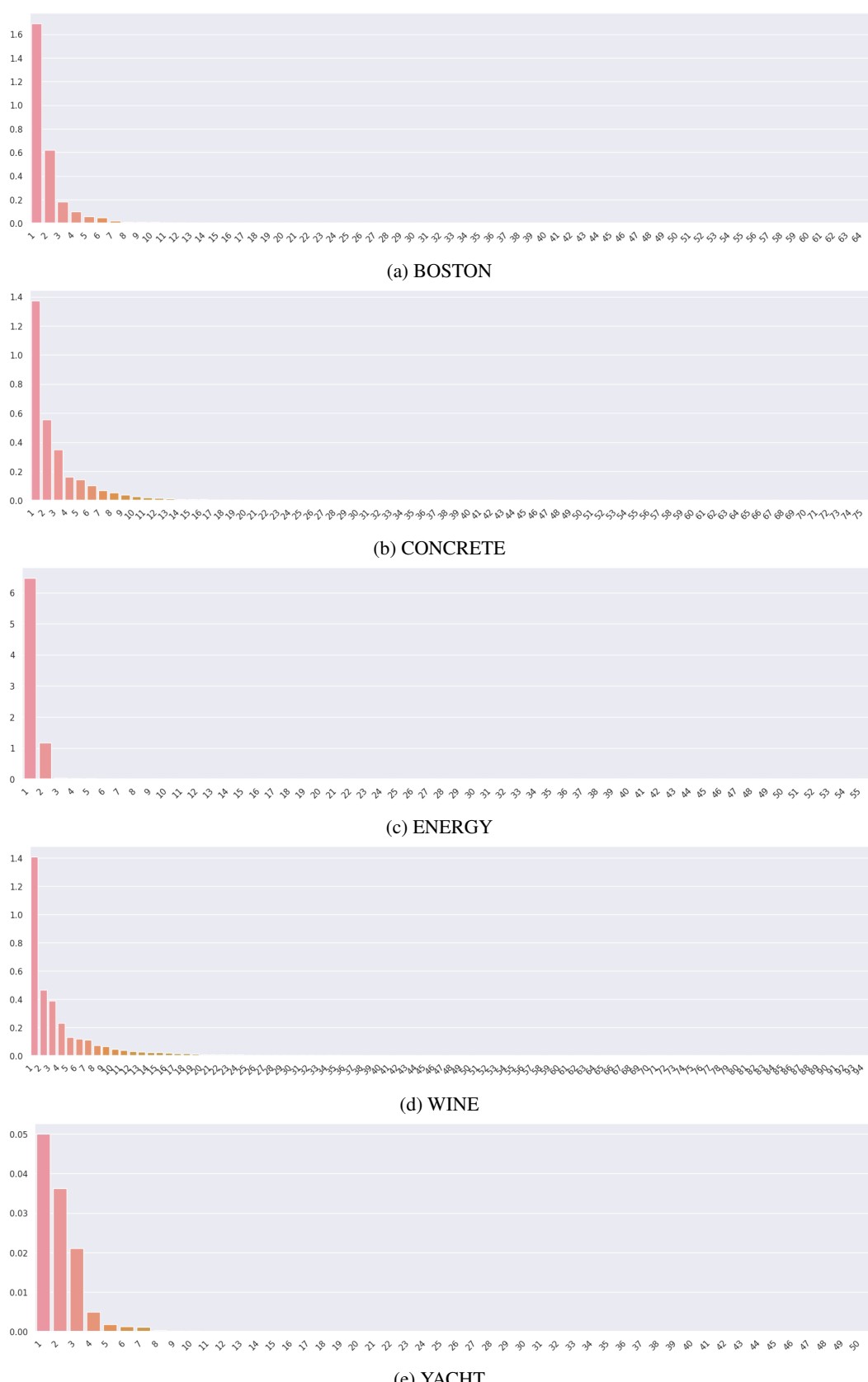

Figure 4: The first 100 positive eigenvalues of $r(X_S, X)k(X, X_S)$ for datasets BOSTON, CON-CRETE, ENERGY, WINE and YACHT.