# OpenReview forum: "Generalized Variational Inference in Function Spaces: Gaussian Measures meet Bayesian Deep Learning"
_NeurIPS.cc/2022/Conference — NeurIPS 2022 Accept_

### Official Review · Reviewer_gB9j · 2022-07-11

**Rating:** 7
**Confidence:** 4
**Soundness:** 3 good
**Presentation:** 3 good
**Contribution:** 3 good

**Summary:**

The paper presents a new methodology to apply variational inference in function space. Starting from the classic ELBO, the authors replace the KL divergence for measures with the Wasserstein distance and, by choosing Gaussian measures for the prior and the approximate posterior, the authors show how to approximate this quantity to have a tractable loss to optimize. Also, the authors present two ways to parameterize the variational posterior, one related to sparse Gaussian processes and the second based on deep neural networks. Finally, the paper is concluded with some experiments, including 1D toy regression, the classic UCI regression benchmark and some image classification problems (with OOD detection).

**Questions:**

- For me, the example shown in Figure 1 is not very illustrative. Fig 1 shows only one of the proposed models with 3 different function estimation problems. For example, it would be interesting to see the comparison between GWI-SVGP and GWI-DNN-SVGP and SVGP alone (SVGP is particularly interesting to see, given that it should be the same model as GWI-SVGP trained on a different loss)
- How are you treating the prior parameters? Are they fixed, cross-validated or optimized? And what about the inducing points? Are these just a sub-sample of the training set or are they optimized?

**Limitations:**

Some additional comments on the limitations would have been welcomed (especially regarding the practical implementation).
No comments on potential negative societal impact, as expected for this kind of work.

**Strengths And Weaknesses:**

## Strengths

- The paper is technically very sound and promises to improve upon standard methods for weight-space and function-space inference of neural networks (and not only).
- The paper is generally easy to follow and it is sufficiently self-contained

## Weaknesses

- The original assumption of working with Gaussian posteriors could be a limiting factor, even if we are working in function-space. For example, a recent work [Pleiss and Cunningham, 2021] on the behaviour of wide parametric (and non) models seems to suggest that the limiting Gaussian posterior performs worse than the BNN/DGP posterior. This is common for all variational inference methods in function space with Gaussian posterior, but I would still appreciate a comment from the Authors on this.
- The experimental evaluation is a bit limited (with nonetheless interesting results). My only concern is on the comparison with other methods, given that those numbers have been copied from various papers. Did you use the same models, same architectures, and same setup?


### Minor
- The reference section can be improved (e.g. miss-capitalizations and arxiv citations when proceedings are available).
- In the experiment, the second parameterization is sometimes referred to as GWI-DNN-SVGP and other times as GWI-Net. Maybe a uniform notation is easier to follow.

Geoff Pleiss, John Patrick Cunningham. The Limitations of Large Width in Neural Networks: A Deep Gaussian Process Perspective. NeurIPS 2021

---

> ### Author Response · Authors · 2022-07-30
> **Rebuttal for Reviewer gB9j**
>
> The authors are very grateful for the careful reading of our manuscript. The comments helped to improve the work significantly.
>
> Weaknesses:
>
> a.) The main goal of the work of Pleiss and Cunningham (2021) appears to be a comparison of the role that depth plays in the BNNs/DGPs. They compare BNNs with increasing width of hidden layers where in the infinite width limit the BNN converges to a GP. Their results seem to suggest that finite width or even the small width regime is beneficial for modeling uncertainty in BNNs/DGPs. It is hard to draw conclusions for our methodology for several reasons. First, we use a neural network architecture for the posterior mean (for a fair comparison) where the width of the hidden layers is given as 10 (cp. l.678-680 in the supplementary material). This means we are for the mean function in the regime described as beneficial in Pleiss and Cunningham (2021). The uncertainty quantification is similar to that of a sparse GP posterior method, which is a consequence of the use of the SVGP kernel. Here the comparison to the uncertainty quantification of a BNN posterior is even more difficult.  Generally speaking, our assumption that the posterior is described by a GP will mostly have two consequences: for a fixed $x$ the posterior distribution $F(x)$ will be unimodal and concentrated around the posterior mean. This unimodal behavior is arguably less problematic in the function space than in the parameter space, but one might still argue that heavier tails than Gaussian could be appropriate for some applications.
>
> b.) The fair comparison aspect is something that we have considered thoroughly. We use the same neural network architecture for all models and similar training procedures. The details are described in the supplementary material (l. 678-680, l. 686-690). We will make this point more explicitly in the main body of the paper.
>
> The minor points will be corrected. Thank you for pointing out these flaws in the citations and terminology.
>
> Questions:
>
> a.) A SVGP method will be added to the plots in Figure 1. Thank you for this great idea. We have added such a plot as well in the rebuttal revision in Appendix A.12, Figure 3.
>
> b.) The prior hyperparameters are chosen by maximizing the marginal log-likelihood on a random subset of $X$ of size $\sqrt N$. Due to the page limit, we had to move these details to the supplementary materials. The details can be found in l.675-l.678. The inducing points are indeed just subsampled randomly from the data points (cf. l. 672- l.674).
>
> Limitations:
>
> Thanks for pointing this out. Indeed after reading the reviews, we decided to include a limitations section in the final manuscript. Here we will discuss limiting assumptions such as Gaussianity as raised by the reviewer and some implementation aspects.

---

> > ### Comment · Reviewer_gB9j · 2022-08-08
> > **Thanks**
> >
> > Thanks for the rebuttal, for answering my questions and for the additional figure. I stand with my previous score, that is I would like to see this paper accepted.

---

### Official Review · Reviewer_6Xua · 2022-07-11

**Rating:** 6
**Confidence:** 4
**Soundness:** 3 good
**Presentation:** 4 excellent
**Contribution:** 3 good

**Summary:**

The authors propose to use the Wasserstein distance between Gaussian measures on infinite-dimensional function spaces to perform generalized variational inference over distributions over functions, with applications to Gaussian processes and Bayesian neural networks.

**Questions:**

Major:

- If we can think of the $X_S$ like inducing points in sparse GPs, it seems like their selection should impact the performance and the estimated distances in the proposed method. Could you comment on how they are chosen? Is there any theoretical guidance on what would be a good choice?
- In the proposed GWInet, it seems that the model ultimately just uses a deterministic NN for the mean function and then wraps a normal GP around that. This seems like a rather complicated method to ultimately "just" perform GP inference. How would one use the proposed method to perform actual BNN inference in function space (that is, using a neural network with stochastic weights), so that one could actually use the learned features to capture the uncertainties instead of having to rely on a GP kernel? This seems crucial, since BNNs outperform GPs in many practical applications.
- State-of-the-art performance on FashionMNIST is 97% and on CIFAR10 around 95%, so the reported performances in Tab. 2 all seem suspiciously low. Could you comment on why that is? I think using baselines that get closer to the performances we actually see in practice would make the comparison stronger and more trustworthy.

Minor:

- l. 75: Regarding variational BNNs in function space, functional SVGD [1] and functional repulsive ensembles [2] seem to be relevant methods to mention.
- Tab. 1: I understand that both SVGP and DNN-SVGP are fit using GWI, right? If so, it seems like a comparison against a standard SVGP implementation would be interesting to see.
- Overall, it's unclear to me whether the proposed method is faster or slower than the baselines. Could you report runtimes (at least rough estimates)?


[1] https://arxiv.org/abs/2106.10760

[2] https://proceedings.neurips.cc/paper/2021/hash/1c63926ebcabda26b5cdb31b5cc91efb-Abstract.html

**Limitations:**

There seems to be no negative societal impact, but the authors could do a slightly better job at addressing the technical limitations, especially regarding runtimes, performance, and how to integrate BNNs (see above).

**Strengths And Weaknesses:**

Strengths:

- The paper is well written and is mathematically rigorous.
- The idea of using Wasserstein inference in function spaces in this way seems novel.
- It is plausible that the proposed method might have advantages over standard variational GPs.

Weaknesses:

- The performance in the experiments seems suspiciously low and a few ablation studies seem to be missing.
- Given that the title advertises "Bayesian Deep Learning", it is unclear how one would exactly apply this to Bayesian neural networks.

---

> ### Author Response · Authors · 2022-07-29
> **Rebuttal for Reviewer 6Xua**
>
> The authors thank the reviewer for pointing out some interesting observations that have helped improve the paper. We will respond below point by point.
>
> Major Questions:
>
> 1. The points $X_S$ are chosen to approximate the spectrum of the appearing covariance operators. In essence, we can approximate the spectrum of a covariance operator by the spectrum of the kernel matrix $k(X,X)$, where $X$ contains all $N$ data points. However, all covariance operators are trace-class operators, which means that the infinitely many eigenvalues have to be summable. This means the sequence of ordered eigenvalues has to converge to zero. Typically this convergence is very fast for kernels such as the SE or Matern kernel.
> Intuition, therefore, suggests that for kernels with quickly decaying eigenvalues, a smaller subset of eigenvalues, for example, the $N_S = 100$ largest ones, may already provide a good approximation to the spectrum of the operator. The $N_S = 100$ largest eigenvalues can be approximated by $k(X_S,X_S)$ where $X_S$ is a randomly chosen subset of $X$ of size $N_S = 100$. We have tried larger sizes of $N_S$ for example $N_S=500$ and seen little change in our results. We also have provided some experimental plots to further prove this point. However, we must admit that we do not have theoretical results to quantify the loss in accuracy. However, the experimental results seem to suggest that we can approximate the spectrum quite well by just choosing a random subset of the data points.
>
> 2. We view our method as providing a novel way of combining the strength of neural networks as function approximators with GP uncertainty quantification. It sits therefore somewhere in between deterministic neural networks and GP inference. The reviewer is right in pointing out that our method does not allow for the standard Bayesian neural network inference, i.e. stochastic weights in a BNN. Rather, our method operates on a function space directly and the posterior is fully specified in terms of its covariance and mean functions. Since our method outperforms several variational inference methods that work explicitly with BNNs (cp. Table 1) it may be fair to ask precisely what aspect of BNNs causes their empirical success. The favourable performance of our method suggests that the function approximation properties of neural networks may be responsible for most of it. The function-space uncertainty quantification akin to that of GPs wrapped around the powerful predictive abilities of the NN as a parametrisation of the posterior mean gives competitive results to BNNs at least when trained with variational inference.
>
> 3. The authors apologize for the confusion. The results in Table 2 are meant to compare different inference procedures for a given architecture, i.e. GVI in function space vs VI in function and weight space. In order to ensure meaningful comparisons with prior related work, we follow the experimental set-up of Immer et. al. (2021) where the same CNN architecture is chosen for all BNN. Our results are state-of-the-art in the sense that for a given NN architecture our inference method performs the best. It would indeed be interesting to consider other more sophisticated NN architectures and explore if GVI objectives lead to further performance and uncertainty quantification gains. We however considered this to be beyond the scope of our paper. We will explain this in more detail in the paper to avoid this confusion.
>
> Minor:
> 1. We will look into these papers and cite them in the related work section.
> 2. Thanks for pointing this out. The comparison with standard SVGP inference is indeed interesting and will be included.
> 3. We give the runtime in l. 258 as $\mathcal{O}(N_S^2 N_B + N_S^3)$. However, as pointed out in the manuscript the full costs are determined by the computations that occur in $r$. Since the evaluation of $r$ is dominated by the inversion of a $M \times M$ matrix where $M = \sqrt N$ our final GWI-net method scales as $N \sqrt N$ (per inference step). This is the same complexity that is typically achieved in the sparse GP literature. We roughly observed that training on a dataset with 40000 datapoints (protein dataset) took 30 minutes on a Nvidia GTX1080 card.  Regarding the baselines: this is a difficult task since most of the authors did not provide a runtime complexity calculation in their method. We hoped that showing that our method is scalable to datasets of the above size without any further problems will provide enough justification for its usefulness. The computational costs of the GWI-net will be explicitly stated in the final version of the manuscript.

---

> > ### Comment · Reviewer_6Xua · 2022-08-04
> > **Thanks**
> >
> > I would like to thank the authors for the detailed rebuttal. It has mostly addressed my concerns and I have thus increased my score.

---

### Official Review · Reviewer_N2sS · 2022-07-11

**Rating:** 7
**Confidence:** 4
**Soundness:** 3 good
**Presentation:** 3 good
**Contribution:** 3 good

**Summary:**

The paper proposes a new probabilistic regression and classification framework called "generalized variational inference in function space" (GVI-FS), which involves taking the standard function-space ELBO formulation and regularizing $\mathbb{E}_\mathbb{Q} [\log p(y|F)]$ with the Wasserstein-2 metric, $\mathcal{W}_2(\mathbb{Q}^F, \mathbb{P}^F)$, instead of the standard KL-divergence, $KL(\mathbb{Q}^F \left| \right| \mathbb{P}^F)$. This substitution is motivated by the generalized variational inference framework of Knoblauch et al. (2019) and the difficulties of working with KL divergences. The paper proposes to use Gaussian measures (GMs) to parameterize the predictive functions, although GPs with additional assumptions provide an equivalent approach. The paper proposes two specific variants of this model, a model in which both the predictive mean and the predictive covariance are parameterized by a sparse variational GP and another in which the predictive mean is parameterized by a neural network and the predictive covariance is parameterized by a sparse variational GP. The paper then presents results from regression, classification, and predictive variance-based out-of-distribution detection, comparing to existing probabilistic methods.

**Questions:**

1. In the experiments section, the performance numbers for baseline models are taken from previous work. Are the model architectures and training procedures for the proposed methods comparable? I didn't notice any notes on this issue.

2. An approximate Wasserstein-2 metric is given (Eq. 15). How good is this approximation in practice? In what situations is it more or less accurate. Some content along these lines would help readers understand the pros and cons of method.

3. (minor) It would be interesting to rely on the GP to influence both the predictive mean and uncertainty in the DNN-SVGP case. This could be achieved by using the NN to output "pseudo-observations" with uncertainty, which are then conditioned using the GP mean and covariance to produce a predictive mean.

**Limitations:**

Yes

**Strengths And Weaknesses:**

Strengths

1. The work is technically sound.

2. The work tackles problems of broad interest: classification, regression, and OOD detection using function-space methods.

3. The work demonstrates substantially improved performance compared to the baseline models tested.


Weaknesses

1. The motivation for the proposed approach in terms of "generalized" variational inference is very weak. Specifically, I find the generalized variational inference framework of Knoblauch et al. (2019) to be so general as to be mostly divorced from the intentions and benefits of Bayesian inference, other than being an explicitly probabilistic method. Here are two alternative interpretations/motivations I find more convincing.

1a) The proposed method is GP regression/classification with Wasserstein-2 regularization. This is very simple and, at least to a first approximation, accurate.

1b) Another connection relates to Bayesian inference more directly. Here is a sketch. Consider the objective

$$\max_Q \mathbb{E}_{F \sim Q} [\log p(y|F) - \log q(F)]  - \lambda \mathcal{W}_2(Q,P)$$

where $\lambda \in \mathbb{R}^+$. This is a close variant of Eq. 12, only differing in the $ \mathbb{E} [- \log q(F)]$ entropy term and the $lambda$ multiplier. With an unrestricted variational family, the optimal approximate posterior in the unregularized $\lambda=0$ case is $q(F) = p(F|y)$, the true posterior.

Consider Langevin diffusion for this unregularized case. We start with a single particle $F_0$ and update in continuous time $t \geq 0$ according to

$$\text{d}F_t = -\nabla V(F_t) \text{d}t + \sqrt{2} \text{d}B_t $$

where $V(\cdot) = - \log p(y|\cdot)$ and $B_t$ denotes Brownian motion. The stationary distribution is $\propto p(F|y) \propto \exp(-V)$, the true posterior. The work of Jordan, Kinderlehrer, & Otto (1998) shows that the marginal law of this diffusion process obeys gradient flow of the functional $KL(\cdot,\pi)$ with respect to the Wasserstein-2 metric.

This implies the resulting marginal law of $F_t$ follows gradient flow of $KL(q(F), p(F|y)) = \mathbb{E}_{F \sim Q} [\log p(y|F) - \log q(F)]$, the first term in the original objective.

Now, back to the original objective, we can interpret $\lambda$ as a Lagrange multiplier, giving the dual formulation

$$\max_Q \mathbb{E}_{F \sim Q} [\log p(y|F) - \log q(F)] \quad \text{s.t. } \mathcal{W}_2(Q,P) \leq C$$

for some $C = C(\lambda) > 0$. this gives the original objective a nice interpretation. If we start the Langevin diffusion process with our variational posterior equal to the prior, $Q_0(F) = P(F)$ and stop the process once the diffusion process has covered a distance of $C$ w.r.t the Wasserstein-2 metric, then we will have approximately optimized our objective. As $C \leftarrow \infty$ (corresponding to $\lambda \leftarrow 0$), the approximate posterior $Q(F)$ approaches $p(F|y)$, the solution to the full Bayesian inference problem. This objective can then be interpreted as Bayesian inference with early stopping.

One wrinkle is that $Q(F)$ is restricted to be a GP. See the recent work of Lambert et al. (2022), "Variational inference via Wasserstein gradient flows" for a good discussion of this issue.

2. The experiments section should contain an additional comparison with a DNN with the same architecture and objective as DNN-SVGP, but with both predictive mean and uncertainty output by the network. This will give the reader a sense of how much the uncertainty handling by the sparse GP is helping performance. An additional comparison without the Wasserstein-2 regularization would also be informative. Additionally, a sparse GP method in addition to the last column in Table 1 (Exact GP) would be helpful for interpreting how good these results are.

3. (minor) l.296-8, VAEs use neural networks to parameterize variational posteriors. I would not consider this approach fundamentally different.

4. (minor) I don't believe $\Sigma$ in Eq. 18 is defined.

5. (minor) The supplied code is almost completely uncommented, making it impractical read.

---

> ### Author Response · Authors · 2022-07-29
> **Rebuttal for Reviewer N2sS**
>
> We thank the first reviewer for the careful reading of our manuscript. The review provided several insights and new perspectives for our paper. A detailed explanation follows below. We respond to the weaknesses in the numerical order provided by the reviewer.
>
> 1. The motivation for our work indeed relies heavily on the rule of three presented in Knoblauch et. all (2018). The fact that our objective function can be interpreted as a regularised measure valued optimisation problem (as mentioned in 1a by the reviewer) is discussed in Knoblauch et. all (2018). We agree with the reviewer that mentioning this explicitly will improve the manuscript. The fact that our objective function can also be interpreted as a Bayesian method with early stopping where the distance is measured by the Wasserstein distance is interesting. We will provide some references in the manuscript to point readers to this connection. It may even prove useful in improving algorithmic designs for Generalised Variational Inference in function space. We will also provide a reference to Lambert (2022) and discuss the limitations of restricting the approximation family to be Gaussian.
>
> 2. Regarding suggestions for additional experiments:
> a) Using neural networks for both m_Q and r is something that we actually tried but was unsuccessful. In essence, the neural network typically does not increase the uncertainty quickly enough once we go away from the observation points. We experimented with several changes in the architecture of the neural network but were not able to obtain satisfying results. This can be easily illustrated with the toy examples, but we also verified it on the UCI regression data set. This discussion will be included in the paper.
> b) A sparse GP and plain NN have been added in the revised version.
>
> Minor Comments will be addressed.
>
> Regarding the Questions:
> 1. Yes, we indeed have taken care to match neural network architecture and training procedures. The details are discussed in the supplementary materials A.7 and A.8.
>
> 2. The approximation quality of the 2-Wasserstein distance is determined by the approximation quality of the spectrum of the corresponding covariance operators. For most kernels used in practice, like SE or Matern kernel, the spectrum decays very quickly, which is why using the first 100 eigenvalues often empirically seems to be sufficient to approximate the spectrum and therefore the 2-Wasserstein distance. We did not manage to derive a theoretical result that gives further insights, but our empirical results suggest a high approximation quality. We have provided some additional plots to substantiate this point in the revision.
>
> 3.  The suggestion to use a neural network to output pseudo-observations indeed sounds intriguing. In the process of writing the paper we considered several ideas along those lines but did not obtain meaningful results. We therefore decided to reserve exploration of this topic for future research.
>
> We would like to thank the reviewer again for their time and insightful comments. We believe they have significantly improved the manuscript.

---

> > ### Comment · Reviewer_N2sS · 2022-08-07
> > **Thanks**
> >
> > I thank the authors for their detailed response. Changes made to address the motivation and model comparisons will substantially improve the manuscript.

---

### Author Response · Authors · 2022-07-31
**General response**

We thank the reviewers for their time and effort. We believe their comments and suggestions have significantly improved the work. We are happy to find that all reviewers find our work rigorous, well-written, and significant. We respond to each reviewer individually below.
The main changes to the manuscript will be as follows:

1. The introduction will mention the interpretations as a regularised optimisation problem over the space of Gaussian measures. We will include references to the connections with Wasserstein flows and Bayesian inference with early stopping.

2. We will discuss the approximation quality of the Wasserstein distance and how it relates to the spectrum of the kernel operators. We have included some plots to illustrate the quick decay of eigenvalues that we observed for the SE kernel.

3. We will describe and contextualize the experiments in more detail. We will mention in the main body that we matched neural network architectures and inference procedures to have a fair comparison between the different models.

4. We will include a comparison to standard SVGP.

5. We will add a section that discusses the limitations of our method. This will include model misspecification, algorithmic aspects, and the limiting nature of the Gaussianity assumption.

---

### Meta-Review · Area_Chair_x8Ui · 2022-08-26

**Recommendation:** Accept
**Confidence:** Certain

**Metareview:**

Technically solid paper that introduces and benchmarks a novel inference framework, with application to inference in GPs. All reviewers recommend to accept, after a decent amount of discussion in which reviewers raised their scores in response to a fairly significant round of updates to the manuscript itself. Recommend to accept, despite some questions regarding overall impact.

**Award:**

No

---

### Decision · Program_Chairs · 2022-09-14

Accept